# Learning Boolean Circuits with Neural Networks

## Abstract

Training neural-networks is computationally hard. However, in practice they are trained efficiently using gradient-based algorithms, achieving remarkable performance on natural data. To bridge this gap, we observe the property of *local correlation*: correlation between small patterns of the input and the target label. We focus on learning deep neural-networks with a variant of gradient-descent, when the target function is a tree-structured Boolean circuit. We show that in this case, the existence of *correlation* between the gates of the circuit and the target label determines whether the optimization succeeds or fails. Using this result, we show that neural-networks can learn the $(\log n)$-parity problem for most product distributions. These results hint that *local correlation* may play an important role in differentiating between distributions that are hard or easy to learn.

## 1 Introduction and Motivation

It is well known (e.g. Livni et al. (2014)) that while deep neural-networks can **express** any function that can be run efficiently on a computer, in the general case, **training** them is computationally hard. Despite this theoretic pessimism, in practice, deep neural networks are successfully trained on real world datasets. Bridging this theoretical-practical gap seems to be the holy grail of theoretical machine learning nowadays. Maybe the most natural direction to bridge this gap is to find a **property** of data distributions that determines whether training is computationally easy or hard. The goal of this paper is to propose such a property.

To motivate this, we first recall the $k$-parity problem: the input is $n$ bits, there is a subset of $k$ relevant bits (which are unknown to the learner), and the output should be 1 if the number of 1's among the relevant bits is even and $-1$ otherwise. It is well known (e.g. Shalev-Shwartz et al. (2017)) that the parity problem can be expressed by a fully connected two layer network or by depth $\log(n)$ locally connected [1] network. We observe the behavior of a one hidden-layer neural network trained on the $k$-parity problem, in two different instances: first, when the underlying distribution is the uniform distribution (i.e. the probability to see every bit is $\frac{1}{2}$); and second, when the underlying distribution is a slightly biased product distribution (the probability for every bit to be 1 is $0.6$). As can be clearly seen in figure 1, adding a slight bias to the probability of each bit dramatically affects the behavior of the network: while on the uniform distribution the training process completely fails, in the biased case it converges to a perfect solution.

This simple experiment shows that a small change in the underlying distribution can cause a dramatic change in the trainability of neural-networks. A key property that differentiates the uniform from the biased distribution is the *correlation* between input bits and the target label. While in the uniform distribution, the correlation between each bit and the label is zero, in the biased case every bit of the $k$ bits in the parity has a non-negligible correlation to the label (we show this formally in section 5). So, *local correlations* between bits of the input and the target label seems to be a promising **property** which separates easy and hard distributions.

In this paper, we analyze the problem of learning tree-structured Boolean circuits with neural-networks. The key property that we assume is having sufficient correlation between every gate in the circuit and the label. We show that a variant of gradient-descent can efficiently learn such

---

[1] i.e. every two adjacent neurons are only connected to one neuron in the upper layer.

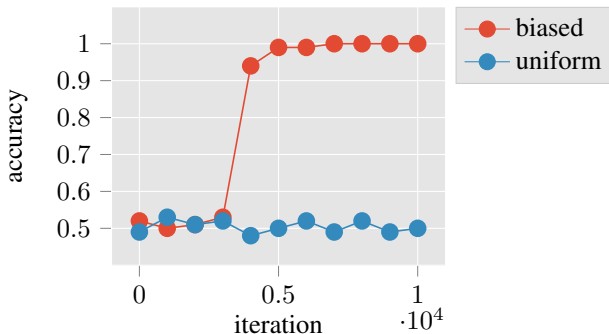

Figure 1: Trainig ReLU networks with one hidden-layer of size 128 with Adam optimizer, on both instances of the $k$-Parity problem ($k = 5, n = 128$). The figure shows the accuracy on a test set.

circuits for some families of distributions, where at the same time, without correlation, gradient-descent is likely to fail. More concretely, we discuss specific target functions and distributions that satisfy the *local correlation* requirement. We show that for most product distributions, gradient-descent learns the $(\log n)$-parity problem (parity on $\log n$ bits of an input with dimension $n$). We further show that for every circuit with AND/OR/NOT gates, there exists a generative distribution, such that gradient-descent recovers the Boolean circuit exactly.

Admittedly, as the primary focus of this paper is on theoretical analysis, the distributions we study are synthetic in nature. However, to explain the empirical success of neural-networks, we need to verify whether the *local correlation* property holds for natural datasets as well. To confirm this, we perform the following simple experiment: we train a network with two hidden-layers on a *single* random patch from images in the ImageNet dataset. We observe that even on a complex task such as ImageNet, a network that gets only a $3 \times 3$ patch as an input, achieves $2.6\%$ top-5 accuracy — much better than a random guess ($0.5\%$ top-5 accuracy). The full results of the experiment are detailed in the appendix. This experiment highlights that, to some extent, natural datasets display a *local correlation* property: even a few "bits" of the input already have some non-negligible information on the target label.

## 2 RELATED WORK

In recent years, the success of neural-networks has inspired an ongoing theoretical research, trying to explain empirical observations about their behavior. Some theoretical works show failure cases of neural-networks. Other works give different guarantees on various learning algorithms for neural-networks. In this section, we cover the main works that are relevant to our paper.

**Failures of gradient-based algorithms**. Various works have shown different examples demonstrating failures of gradient-based algorithm. The work of Shamir (2018) shows failures of gradient descent, both in learning natural target functions and in learning natural distributions. The work of Shalev-Shwartz et al. (2017) shows that gradient-descent fails to learn parities and linear-periodic functions under the uniform distribution. In Das et al. (2019), a hardness result for learning random deep networks is shown. Other similar failure cases are also covered in Abbe & Sandon (2018); Malach & Shalev-Shwartz (2019). While the details of these works differ, they all share the same key principal - if there is no local correlation, gradient-descent fails. Our work complements these results, showing that in some cases, when there are local correlations to the target, gradient-descent succeeds to learn the target function.

**Learning neural-networks with gradient-descent**. Recently, a large number of papers have provided positive results on learning neural-networks with gradient-descent. Generally speaking, most of these works show that over-parametrized neural-networks, deep or shallow, achieve performance that is competitive with kernel-SVM. Daniely (2017) shows that SGD learns the conjugate kernel associated with the architecture of the network, for a wide enough neural-network. The work of Brutzkus et al. (2017) shows that SGD learns a neural-network with good generalization, when the target function is linear. A growing number of works show that for a specific kernel induced

by the network activation, called the Neural Tangent Kernel (NTK), gradient-descent learns over-parametrized networks, for target functions with small norm in the reproducing kernel Hilbert space (see the works of Jacot et al. (2018); Xie et al. (2016); Oymak & Soltanolkotabi (2018); Allen-Zhu et al. (2018a;b); Oymak & Soltanolkotabi (2019); Arora et al. (2019); Du et al. (2018); Ma et al. (2019); Lee et al. (2019)). While these results show that learning neural-networks with gradient-descent is not hopeless, they are in some sense disappointing — in practice, neural-networks achieve performance that are far better than SVM, a fact that is not explained by these works. A few results do discuss success cases of gradient-descent that go beyond the kernel-based analysis (Brutzkus & Globerson, 2017; 2019; Allen-Zhu & Li, 2019; Yehudai & Shamir, 2019). However, these works still focus on very simple cases, such as learning a single neuron, or learning shallow neural-networks in restricted settings. In this work we deal with learning deep networks, going beyond the common reduction to linear classes of functions.

**Layerwise optimization algorithms**. In this paper, we analyze the behavior of layerwise gradient-descent — optimizing one layer at a time, instead of the common practice to optimize the full network end-to-end. We do so since such algorithm greatly simplifies our theoretical analysis. While layerwise training is not a common practice, recent works (Belilovsky et al., 2018; 2019) have shown that such algorithms achieve performance that are competitive with the standard end-to-end approach, scaling up to the ImageNet dataset. We note that other theoretical works have studied iterative algorithms that learn neural-networks layer-by-layer (Arora et al., 2014; Malach & Shalev-Shwartz, 2018). However, our work focuses specifically on layerwise gradient-descent, considering the problem of learning Boolean circuits.

**Learning Boolean Circuits**. The problem of learning Boolean circuits has been studied in the classical literature of theoretical machine learning. The work of Kearns et al. (1987) gives various positive and negative results on the learnability of Boolean Formulas, including Boolean circuits. The work of Linial et al. (1989) introduces an algorithm that learns a constant-depth circuit in quasi-polynomial time. Another work by Kalai (2018) discusses various properties of learning Boolean formulas and Boolean circuits. Our work differs from the above in various aspects. Our main focus is learning deep neural-networks with gradient descent, where the target function is implemented by a Boolean circuit, and we do not aim to study the learnability of Boolean circuits in general. Furthermore, we consider Boolean circuits where a gate can take any Boolean functions, and not only AND/OR/NOT, as is often considered in the literature of Boolean circuits. On the other hand, we restrict ourselves to the problem of learning circuits with a fixed structure of full binary trees. We are not aware of any work studying a problem similar to ours.

## 3    PROBLEM SETTING

We consider the problem of learning binary classification functions over the Boolean cube. So, let $\mathcal{X} = \{\pm 1\}^n$ be the instance space and $\mathcal{Y} = \{\pm 1\}$ be the label set. Throughout the paper, we assume the target function is given by a Boolean circuit. In general, such assumption effectively does not limit the set of target functions, as any computable function can be implemented by a Boolean circuit. We define a circuit $C$ to be a directed graph with $n$ input nodes and a single output node, where each inner node has exactly two incoming edges, and is labeled by some arbitrary Boolean function $f : \{\pm 1\}^2 \to \{\pm 1\}$, which we call a gate [2]. For each node $v$ in $C$ we denote by $\gamma(v) \in \{f : \{\pm 1\}^2 \to \{\pm 1\}\}$ its gate. We recursively define $h_{v,C} : \{\pm 1\}^n \to \{\pm 1\}$ to be:

$$h_{v,C}(\boldsymbol{x}) = \gamma(v) \left( h_{u_1,C}(\boldsymbol{x}), h_{u_2,C}(\boldsymbol{x}) \right)$$

where $u_1, u_2$ are the two nodes with outcoming edges to $v$. Finally, define $h_C = h_{o,C}$, where $o$ is the output node.

We study the problem of learning the target function $h_C$, when $C$ is a full binary tree, and $n = 2^d$, where $d$ is the depth of the tree. The leaves of the tree are the input bits, ordered by $x_1, \ldots x_n$. Admittedly, such assumption greatly limits the set of target functions, but still gives a rather rich family of functions. For example, such circuit can calculate the parity function on any $k$ bits of the input (the function calculated by $f(\boldsymbol{x}) = \prod_{i \in I} x_i$ for some set of indexes $I$). We note that the total number of functions calculated by such tree grows like $6^n$, as shown in Farhoodi et al. (2019).

---

[2]Note that in the literature on Boolean circuits it is often assumed that the gates are limited to being AND/OR and NOT. We allow the gates to take any Boolean function, which makes this model somewhat stronger.

We introduce a few notations that are used in the sequel. Fix some tree structured binary circuit $C$. This circuit has $d$ levels, and we denote $v_{i,j}$ the $j$-th node in the $i$-th level of the tree, and denote $\gamma_{i,j} = \gamma(v_{i,j})$. Fix some $i \in [d]$, let $n_i := 2^i$, and denote by $\Gamma_i : \{\pm 1\}^{n_i} \to \{\pm 1\}^{n_i/2}$ the function calculated by the $i$-th level of the circuit:

$$\Gamma_i(\boldsymbol{x}) = \big(\gamma_{i-1,1}(x_1, x_2), \ldots, \gamma_{i-1,n_i/2}(x_{n_i-1}, x_{n_i})\big)$$

For $i < i'$, we denote: $\Gamma_{i\ldots i'} := \Gamma_i \circ \cdots \circ \Gamma_{i'}$. So, the full circuit is given by $h_C(\boldsymbol{x}) = \Gamma_{1\ldots d}(\boldsymbol{x})$.

As noted, our goal is to learn Boolean circuits with neural-networks. To do so, we use a network architecture that aims to imitate the Boolean circuits described above. We replace each Boolean gate with a *neural-gate*: a one hidden-layer ReLU network, with a hard-tanh[3] activation on its output. Formally, let $\sigma$ be the ReLU activation, and let $\phi$ be the hard-tanh activation, so:

$$\sigma(x) = \max(x, 0), \ \phi(x) = \begin{cases} -1 & x \le -1 \\ x & x \in (-1, 1) \\ 1 & x \ge 1 \end{cases}$$

Define a *neural-gate* to be a neural-network with one hidden layer, input dimension 2, with ReLU activation for the hidden-layer and hard-tanh for the output node. Namely, denote $g_{\boldsymbol{w},\boldsymbol{v}} : \mathbb{R}^2 \to \mathbb{R}$ such that:

$$g_{\boldsymbol{w},\boldsymbol{v}}(\boldsymbol{x}) = \phi(\sum_{l=1}^k v_i \sigma(\langle \boldsymbol{w}_l, \boldsymbol{x}\rangle))$$

Notice that a *neural-gate* $g_{\boldsymbol{w},\boldsymbol{v}}$ of width 4 or more can implement any Boolean gate. That is, we can replace any Boolean gate with a neural-gate, and maintain the same expressive power. To implement the full Boolean circuit defined above, we construct a deep network of depth $d$ (the depth of the Boolean circuit), with the same structure as the Boolean circuit. We define $d$ blocks, each block has *neural-gates* with the same structure and connectivity as the Boolean circuit. A block $B_{\boldsymbol{W}^{(i)},\boldsymbol{V}^{(i)}} : \mathbb{R}^{2^i} \to \mathbb{R}^{2^{i-1}}$, is defined by:

$$B_{\boldsymbol{W}^{(i)},\boldsymbol{V}^{(i)}}(\boldsymbol{x}) = [g_{\boldsymbol{w}^{(i,1)},\boldsymbol{v}^{(i,1)}}(x_1, x_2), g_{\boldsymbol{w}^{(i,2)},\boldsymbol{v}^{(i,2)}}(x_3, x_4), \ldots, g_{\boldsymbol{w}^{(i,2^{i-1})},\boldsymbol{v}^{(i,2^{i-1})}}(x_{2^i-1}, x_{2^i})]$$

We consider the process of training neural-networks of the form $\mathcal{N}_{\boldsymbol{\mathsf{W}},\boldsymbol{\mathsf{V}}} = B_{\boldsymbol{W}^{(1)},\boldsymbol{V}^{(1)}} \circ \cdots \circ B_{\boldsymbol{W}^{(d)},\boldsymbol{V}^{(d)}}$. Notice that indeed, a network $\mathcal{N}_{\boldsymbol{\mathsf{W}},\boldsymbol{\mathsf{V}}}$ can implement any tree-structured Boolean circuit of depth $d$. In practice, neural-networks are trained with gradient-based optimization algorithm, in an end-to-end fashion. That is, the weights of all the layers are optimized together, with gradient updates on a given sample. To simplify the analysis, we instead consider a layerwise optimization algorithm, that performs gradient updates layer-by-layer. While this approach is much less popular, it has been recently shown to achieve performance that are comparable to end-to-end training, scaling up to the ImageNet dataset (Belilovsky et al., 2018).

Denote by $P$ the average-pooling operator, defined by $P(x_1, \ldots, x_n) = \frac{1}{n}\sum_{i=1}^n x_i$. Denote the hinge-loss by $\ell(\hat{y}, y) = \max(1 - y\hat{y}, 0)$ and denote the loss on the distribution by $L_{\mathcal{D}}(f) = \mathbb{E}_{(\boldsymbol{x},y)\sim\mathcal{D}}[\ell(f(\boldsymbol{x}), y)]$. For a sample $S \subseteq \mathcal{X} \times \mathcal{Y}$, denote the loss on the sample by $L_S(f) = \frac{1}{|S|}\sum_{(\boldsymbol{x},y)\in S} \ell(f(\boldsymbol{x}), y)$. The layerwise gradient-descent algorithm for learning deep networks is described in algorithm 1.

For simplicity, we assume that the second layer of every *neural-gate* is fixed, such that $v \in \{\pm 1\}$. Notice that this does not limit the expressive power of the network. Algorithm 1 iteratively optimizes the output of the network's layers, starting from the bottom-most layer. For each layer, the average-pooling operator is applied to reduce the output of the layer to a single bit, and this output is optimized with respect to the target label. Note that in fact, we can equivalently optimize each *neural-gate* separately and achieve the same algorithm. However, we present a layerwise training process to conform with algorithms used in practice.

---

[3] We chose to use the hard-tanh activation over the more popular tanh activation since it simplifies our theoretical analysis. However, we believe the same results can be given for the tanh activation.

---

**Algorithm 1** Layerwise Gradient-Descent

---

**input**:
    Sample $S \subseteq \mathcal{X} \times \mathcal{Y}$, number of iterations $T \in \mathbb{N}$, learning rate $\eta \in \mathbb{R}$.
Let $\mathcal{N}_d \leftarrow id$
**for** $i = d \dots 1$ **do**
    Initialize $\boldsymbol{W}_0^{(i)}, \boldsymbol{V}_0^{(i)}$.
    **for** $t = 1 \dots T$ **do**
        Update $\boldsymbol{W}_t^{(i)} \leftarrow \boldsymbol{W}_{t-1}^{(i)} - \eta \frac{\partial}{\partial \boldsymbol{W}_{t-1}^{(i)}} L_S(P(B_{\boldsymbol{W}_{t-1}^{(i)}, \boldsymbol{V}_0^{(i)}} \circ \mathcal{N}_i))$
    **end for**
    Update $\mathcal{N}_{i-1} \leftarrow B_{\boldsymbol{W}_T^{(i)}, \boldsymbol{V}_0^{(i)}} \circ \mathcal{N}_i$
**end for**
Return $\mathcal{N}_0$

---

## 4 MAIN RESULTS

Our main result shows that algorithm 1 can learn a function implemented by the circuit $C$, when running on "nice" distributions, with the local correlation property. We start by describing the distributional assumptions needed for our main results. Let $\mathcal{D}$ be some distribution over $\mathcal{X} \times \mathcal{Y}$. For some function $f : \mathcal{X} \rightarrow \mathcal{X}'$, we denote by $f(\mathcal{D})$ the distribution of $(f(x), y)$ where $(x, y) \sim \mathcal{D}$. Let $\mathcal{D}^{(i)}$ be the distribution $\Gamma_{(i+1)\dots d}(\mathcal{D})$. Denote by $c_{i,j}$ the correlation between the output of the $j$-th gate in the $i$-th layer and the label, so: $c_{i,j} := \mathbb{E}_{\mathcal{D}^{(i)}}[x_j y]$.

Define the influence of the $j$-th gate in the $i$-th layer with respect to the uniform distribution $(U)$ by:

$$\mathcal{I}_{i,j} := \mathbb{P}_{\boldsymbol{x} \sim U}[\Gamma_{i-1}(\boldsymbol{x}) \neq \Gamma_{i-1}(\boldsymbol{x} \oplus e_j)] := \mathbb{P}_{\boldsymbol{x} \sim U}[\Gamma_{i-1}(\boldsymbol{x}) \neq \Gamma_{i-1}(x_1, \dots, -x_j, \dots, x_n)]$$

Now, we introduce the main assumption on the distribution $\mathcal{D}$. We assume the following:

**Assumption 1.** *(local correlation) There exists some* $\Delta \in (0,1)$ *such that for every layer* $i \in [d]$ *and for every gate* $j \in [2^i]$ *with* $\mathcal{I}_{i,j} \neq 0$*, the value of* $c_{i,j}$ *satisfies* $|c_{i,j}| > |\mathbb{E}_{\mathcal{D}}[y]| + \Delta$.

Another technical assumption we need to make is the following:

**Assumption 2.** *(label bias) There exists some* $\beta \in (0,1)$ *such that* $|\mathbb{E}_{\mathcal{D}}[y]| > \beta$.

Before we present the main results, we wish to discuss the distributional assumptions given above. Assumption 1 is the key assumption required for the algorithm to succeed in learning the target function. Essentially, this assumption requires that the output of every gate in the circuit will "explain" the label slightly better then simply observing the bias between positive and negative examples. Clearly, gates that have no influence on the target function never satisfy this property, so we require it only for influencing gates. While this is a strong assumption, in section 5 we discuss examples of distributions where this assumption typically holds. Furthermore, the experiment described in section 1 hints that this assumption may hold for natural data. Assumption 2 is a simple technical assumption, that requires that the distribution of positive and negative examples is slightly biased. In a sense, we expect that "most" distributions would not be exactly balanced, so this assumption is easy to satisfy.

Now, consider the case where $\mathcal{D}$ (limited to $\mathcal{X}$) is a product distribution: for every $j \neq j'$, the variables $x_j$ and $x_{j'}$ are independent, for $(\boldsymbol{x}, y) \sim \mathcal{D}$. A simple argument shows that any product distribution $\mathcal{D}$ that satisfies assumptions 1, satisfies the following properties:

**Property 1.** *There exists some* $\Delta \in (0,1)$ *such that for every layer* $i \in [d]$ *and for every gate* $j \in [2^i]$*, the output of the* $j$-th *gate in the* $i$-th *layer satisfies one of the following:*

- *The value of the gate* $j$ *is independent of the label* $y$*, and its influence is zero:* $\mathcal{I}_{i,j} = 0$.

- *The value of* $c_{i,j}$ *satisfies* $|c_{i,j}| > |\mathbb{E}_{\mathcal{D}}[y]| + \Delta$.

**Property 2.** *For every layer* $i \in [d]$*, and for every gate* $j \in [2^{i-1}]$*, the value of* $(x_{2j-1}, x_{2j})$ *(i.e, the input to the* $j$-th *gate of layer* $i - 1$*) is independent of the label* $y$ *given the output of the* $j$-th *gate:*

$$\mathbb{P}_{(x,y) \sim \mathcal{D}^{(i)}}[(x_{2j-1}, x_{2j}) = \boldsymbol{p}, y = y'|\gamma_{i-1,j}(x_{2j-1}, x_{2j})]$$
$$= \mathbb{P}_{(\boldsymbol{x},y) \sim \mathcal{D}^{(i)}}[(x_{2j-1}, x_{2j}) = \boldsymbol{p}|\gamma_{i-1,j}(x_{2j-1}, x_{2j})] \cdot \mathbb{P}_{(\boldsymbol{x},y) \sim \mathcal{D}^{(i)}}[y = y'|\gamma_{i-1,j}(x_{2j-1}, x_{2j})]$$

Property 1 is immediate from assumption 1. The following lemma shows that property 2 is satisfied as well for any product distribution:

**Lemma 1.** *Assume $\mathcal{D}$ (restricted to $\mathcal{X}$) is a product distribution (i.e., for every $j \neq j'$ we have that $x_j$ and $x_{j'}$ are independent, for $(\boldsymbol{x}, y) \sim \mathcal{D}$). Then $\mathcal{D}$ satisfies property 2.*

Notice that properties 1, 2 may hold for distributions that are not product distribution (as we show in the next section). Specifically, property 2 is a very common assumption in the field of Graphical Models (see Koller & Friedman (2009)). For our results to hold in a more general setting, we use properties 1 and 2, instead of assuming that $\mathcal{D}$ is a product distribution satisfying assumption 1. So, given a distribution satisfying properties 1, 2 and assumption 2, we show that algorithm 1 achieves an arbitrarily good approximation with high probability, with sample complexity and run-time quasi-polynomial in the dimension $n$:

**Theorem 1.** *Let $\mathcal{D}$ be a distribution satisfying properties 1, 2 and assumption 2. Assume that for every $i$ we initialize $\boldsymbol{W}_0^{(i)}$ such that $\left\| \boldsymbol{W}_0^{(i)} \right\|_{\max} \leq \frac{1}{4\sqrt{2}k}$. Fix some $\epsilon, \delta > 0$ and assume that $k \geq \log^{-1}(\frac{4}{3}) \log(\frac{2nd}{\delta})$, and that $\eta \leq \frac{1}{16k}$. Assume we sample $S \sim \mathcal{D}$, with $|S| > \frac{128}{\epsilon^2 \min\{\Delta, 2\beta\}^2} n^{11+4\log n - 2\log \min\{\Delta, 2\beta\}} \log(\frac{8nd}{\delta})$. Then, with probability at least $1 - \delta$, when running algorithm 1 on the sample $S$, the algorithm returns the a function such that:*

$$\mathbb{E}_{(\boldsymbol{x}, y) \sim \mathcal{D}} [\mathcal{N}_0(\boldsymbol{x}) \neq h_C(\boldsymbol{x})] \leq \epsilon$$

*when running $T > \frac{3\sqrt{2}}{\eta \min\{\Delta, 2\beta\}\epsilon} n^{6.5 + 2\log n - \log \min\{\Delta, 2\beta\}}$ steps for each layer.*

The above shows a learnability result in the standard PAC setting (given our distributional assumptions), where we only guarantee approximation of the target function under the given distribution. In fact, we can get a stronger result, and show that the algorithm learns the function $h_C$ exactly, with run-time and sample complexity polynomial in $n$. To get this result, we need to require that there is no feasible pattern (pair of bits) in the Boolean circuit that is extremely rare:

**Assumption 3.** *There exists some $\epsilon \in (0, 1)$ such that for every layer $i \in [d]$, for every gate $j \in [2^{i-1}]$ and for every $\boldsymbol{p} \in \{\pm 1\}^2$ such that $\mathbb{P}_{(\boldsymbol{x}, y) \sim \mathcal{D}^{(i)}} [(x_{2j-1}, x_{2j}) = \boldsymbol{p}] \neq 0$, it holds that: $\mathbb{P}_{(\boldsymbol{x}, y) \sim \mathcal{D}^{(i)}} [(x_{2j-1}, x_{2j}) = \boldsymbol{p}] \geq \epsilon$.*

In section 5 we discuss distributions that satisfies assumption 3. Given all the above assumptions, we get the following:

**Theorem 2.** *Let $\mathcal{D}$ be a distribution satisfying properties 1, 2 and assumptions 2, 3. Assume that for every $i$ we initialize $\boldsymbol{W}_0^{(i)}$ such that $\left\| \boldsymbol{W}_0^{(i)} \right\|_{\max} \leq \frac{1}{4\sqrt{2}k}$. Fix some $\delta > 0$ and assume that $k \geq \log^{-1}(\frac{4}{3}) \log(\frac{2nd}{\delta})$, and that $\eta \leq \frac{1}{16k}$. Assume we sample $S \sim \mathcal{D}$, with $|S| > \frac{128}{\epsilon^2 \min\{\Delta, 2\beta\}^2} \log(\frac{8nd}{\delta})$. Then, with probability at least $1 - \delta$, when running algorithm 1 on the sample $S$, the algorithm returns a function such that $\mathcal{N}_0(\boldsymbol{x}) = h_C(\boldsymbol{x})$ for all $\boldsymbol{x} \in \mathcal{X}$, when running $T > \frac{3\sqrt{2}n}{\eta \min\{\Delta, 2\beta\}\epsilon}$ steps for each layer.*

We give the full proof of the theorems in the appendix, and give a sketch of the argument here. Observe that the input to the $(i, j)$-th *neural-gate* is a pattern of two bits. The target gate (the $(i, j)$-th gate in the circuit $C$) identifies each of the four possible patterns with a single output bit. For example, if the gate is OR, then the patterns $\{(1, 1), (-1, 1), (1, -1)\}$ get the value 1, and the pattern $(-1, -1)$ gets the value $-1$. Fix some pattern $\boldsymbol{p} \in \{\pm 1\}^2$, and assume that the output of the $(i, j)$-th gate on the pattern $\boldsymbol{p}$ is 1. Since we assume the output of the gate is correlated with the label, the loss function draws the output of the *neural-gate* on the pattern $\boldsymbol{p}$ toward the *correlation* of the gate. In the case where the output of the gate on $\boldsymbol{p}$ is $-1$, the output of the *neural-gate* is drawn to the opposite sign of the *correlation*. All in all, the optimization separates the patterns that evaluate to 1 from the patterns that evaluate to $-1$. In other words, the *neural-gate* learns to implement the target gate. This way, we can show that the optimization process makes the network recover all the influencing gates, so that at the end of the optimization the network implements the circuit.

Observe that when there is no correlation, the above argument fails immediately. Since the label is slightly biased, when there is no correlation the output of the *neural-gate* is drawn towards the bias of the label for all the input patterns, regardless of the value of the gate. If the gate is not influencing

the target function (i.e. $\mathcal{I}_{i,j} = 0$), then this clearly doesn't effect the overall behavior. However, if there exists some influencing gate with no correlation to the label, then the output of the *neural-gate* will be constant on all its input patterns. Hence, the algorithm will fail to learn the target function. This shows that assumption 1 is in fact critical for the success of the algorithm.

## 5 DISTRIBUTIONS

In the previous section we showed that algorithm 1 can learn tree-structured Boolean circuits in polynomial run-time and sample complexity. These results require some non-trivial distributional assumptions. In this section we study specific families of distributions, and show that they satisfy the above assumptions.

First, we study the problem of learning a parity function on $\log n$ bits of the input, when the underlying distribution is a product distribution. The problem of learning parities was studied extensively in the literature of machine learning theory (Feldman et al., 2006; 2009; Blum et al., 2003; Shalev-Shwartz et al., 2017; Brutzkus et al., 2019), and serves as a good case-study for the above results. In the $(\log n)$-parity problem, we show that in fact *most* product distributions satisfy assumptions 1-3, hence our results apply to most product distributions. Next, we study distributions given by a generative model. We show that for every circuit with gates AND/OR and NOT, there exists a distribution that satisfies the above assumptions, so algorithm 1 can learn any such circuit exactly.

### 5.1 PRODUCT DISTRIBUTIONS

We observe the $k$-Parity problem, where the target function is $f(\boldsymbol{x}) = \prod_{j \in I} x_j$ some subset $I \subseteq [n]$ of size $|I| = k$. A simple construction shows that $f$ can be implemented by a tree structured circuit as defined previously. We define the gates of the first layer by:

$$\gamma_{d-1,j}(z_1, z_2) = \begin{cases} z_1 z_2 & x_{2j-1}, x_{2j} \in I \\ z_1 & x_{2j-1} \in I, x_{2j} \notin I \\ z_2 & x_{2j} \in I, x_{2j-1} \notin I \\ 1 & o.w \end{cases}$$

And for all other layers $i < d - 1$, we define: $\gamma_{i,j}(z_1, z_1) = z_1 z_2$. Then we get the following:

**Lemma 2.** *Let $C$ be a Boolean circuit as defined above. Then: $h_C(\boldsymbol{x}) = \prod_{j \in I} x_j = f(\boldsymbol{x})$.*

Now, let $\mathcal{D}_{\mathcal{X}}$ be some product distribution over $\mathcal{X}$, and denote $p_j := \mathbb{P}_{\mathcal{D}_{\mathcal{X}}}[x_j = 1]$. Let $\mathcal{D}$ be the distribution of $(\boldsymbol{x}, f(\boldsymbol{x}))$ where $\boldsymbol{x} \sim \mathcal{D}_{\mathcal{X}}$. Then for the circuit defined above we get the following result:

**Lemma 3.** *Fix some $\xi \in (0, \frac{1}{4})$. For every product distribution $\mathcal{D}$ with $p_j \in (\xi, \frac{1}{2}-\xi) \cup (\frac{1}{2}+\xi, 1-\xi)$ for every $j$, it holds that if $\mathcal{I}_{i,j} \neq 0$ then $|c_{i,j}| - |\mathbb{E}[y]| \geq \xi^k$ and $\mathbb{P}_{(\boldsymbol{z},y) \sim \Gamma_{(i+1)...d}(\mathcal{D})}[z_j = 1] \in (\xi, 1-\xi)$.*

The above lemma shows that every product distribution that is far enough from the uniform distribution, or from a constant distribution, satisfies assumptions 1 and 2 with $\beta, \Delta = (2\xi)^k$. Using the fact that at each layer, the output of each gate is an independent random variable (since the input distribution is a product distribution), we get that assumption 3 is satisfied with $\epsilon = \xi^2$. This gives us the following result:

**Corollary 1.** *Let $\mathcal{D}$ be a product distribution with $p_j \in (\xi, \frac{1}{2} - \xi) \cup (\frac{1}{2} + \xi, 1 - \xi)$ for every $j$, with the target function being the ($\log n$)-Parity (i.e., $k = \log n$). Then, when running algorithm 1 as described in Theorem 2, with probability $1 - \delta$ the algorithm returns the true target function $h_C$, with run-time and sample complexity polynomial in $n$.*

### 5.2 GENERATIVE MODELS

Next, we move beyond product distributions, and observe families of distributions given by a generative model. We limit ourselves to circuits where each gate is chosen from the set $\{\wedge, \vee, \neg\wedge, \neg\vee\}$. For every such circuit, we define a generative distribution as follows: we start by sampling a label

for the example, from a slightly imbalanced distribution (to satisfy assumption 2). Then iteratively, for every gate, we sample uniformly at random a pattern from all the pattern that give the correct output. For example, if the label is 1 and the topmost gate is OR, we sample a pattern uniformly from $\{(1,1),(1,-1),(-1,1)\}$. The sampled pattern determines what should be the output of the second topmost layer. For every gate in this layer, we sample again a pattern that will result in the correct output. We continue in this fashion until reaching the bottom-most layer, which defines the observed example.

Formally, for a given gate $\Gamma \in \{\wedge, \vee, \neg\wedge, \neg\vee\}$, we denote the following sets of patterns:
$$S_\Gamma = \{\boldsymbol{v} \in \{\pm1\}^2 \ : \ \Gamma(v_1, v_2) = 1\}, \ S_\Gamma^c = \{\pm1\}^2 \setminus S_\Gamma$$
We recursively define $\mathcal{D}^{(0)}, \ldots, \mathcal{D}^{(d)}$, where $\mathcal{D}^{(i)}$ is a distribution over $\{\pm1\}^{2^i} \times \{\pm1\}$:

- $\mathcal{D}^{(0)}$ is a distribution supported on $\{(1,1),(-1,-1)\}$ such that $\mathbb{P}_{\mathcal{D}^{(0)}}[(1,1)] = \frac{1}{2} + \xi$ and $\mathbb{P}_{\mathcal{D}^{(0)}}[(-1,-1)] = \frac{1}{2} - \xi$, for some $0 < \xi < \frac{1}{12}\left(\frac{2}{3}\right)^d$.
- To sample $(\boldsymbol{x}, y) \sim \mathcal{D}^{(i)}$, first sample $(\boldsymbol{z}, y) \sim \mathcal{D}^{(i-1)}$. Then, for all $j \in [2^{i-1}]$, if $z_j = 1$ sample $\boldsymbol{x}'_j \sim Uni(S_{\gamma_{i,j}})$, and if $z_j = -1$ sample $\boldsymbol{x}'_j \sim Uni(S^c_{\gamma_{i,j}})$. Set $\boldsymbol{x} = [\boldsymbol{x}'_1, \ldots, \boldsymbol{x}'_{2^{i-1}}] \in \{\pm1\}^{2^i}$, and return $(\boldsymbol{x}, y)$.

Then we have the following results:

**Lemma 4.** *For every $i \in [d]$ and every $j \in [2^i]$, denote $c_{i,j} = \mathbb{E}_{(\boldsymbol{x},y)\sim\mathcal{D}^{(i)}}[x_j y]$. Then we have:*
$$|c_{i,j}| - |\mathbb{E}[y]| > \frac{1}{3}\left(\frac{2}{3}\right)^d = \frac{1}{3}n^{\log(2/3)}$$

We also need the following simple observation:

**Lemma 5.** *For every $i \in [d]$ we have $\Gamma_i(\mathcal{D}^{(i)}) = \mathcal{D}^{(i-1)}$.*

By definition, we have $\mathbb{E}[y] = 2\xi$, so $\mathcal{D}^{(d)}$ satisfies assumption 2 with $\beta = \xi$. Notice that from Lemma 4, the distribution $\mathcal{D}^{(d)}$ satisfies property 1 with $\Delta = \frac{1}{3}n^{\log(2/3)}$ (note that since we restrict the gates to AND/OR/NOT, all gates have influence). By its construction, the distribution also satisfies property 2, and it satisfies assumption 3 with $\epsilon = \left(\frac{1}{4}\right)^d = \frac{1}{n^2}$. Therefore, we can apply Theorem 2 on the distribution $\mathcal{D}^{(d)}$, and get that algorithm 1 learns the circuit $C$ *exactly* in polynomial time. This leads to the following nice corollary:

**Corollary 2.** *With the assumptions and notations of Theorem 2, for every circuit $C$ with gates in $\{\wedge, \vee, \neg\wedge, \neg\vee\}$, there exists a distribution $\mathcal{D}$ such that when running algorithm 1 on a sample from $\mathcal{D}$, the algorithm returns $h_C$ with probability $1 - \delta$, in polynomial run-time and sample complexity.*

Note that the fact that for every circuit there exists a distribution that can be learned in the PAC setting is trivial: simply take a distribution that is concentrated on a single positive example, and approximating the target function on such distribution is achieved by a classifier that always returns a positive prediction. However, showing that there exists a distribution on which algorithm 1 *exactly* recovers the circuit, is certainly non-trivial.

## 6 DISCUSSION

In this paper we suggested the property of *local corrleation* as a possible candidate for differentiating between hard and easy distributions. We showed that on the task of learning tree-structured Boolean circuits, the existence of *local correlations* between the gates and the target label allows layerwise gradient-descent to learn the target circuit. Furthermore, we showed specific tasks and distributions which satisfy the *local correlation* property. These results raise a few open questions, which we leave for future work. The most immediate research problem is showing similar results for more general structures of Boolean circuit, and on a wider range of distributions (beyond product distributions or generative models). More generally, we suggest that the *local correlation* property may be important in a broader context, beyond Boolean circuits. For example, examining whether an equivalent property exists when the target function is a convolutional network is an extremely interesting open problem. Needless to say, finding other properties of natural distribution that determine whether gradient-based algorithms succeed or fail is another promising research direction.

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

## A    EXPERIMENTS

Figure 2 details the results of the ImageNet experiment discussed in the introduction.

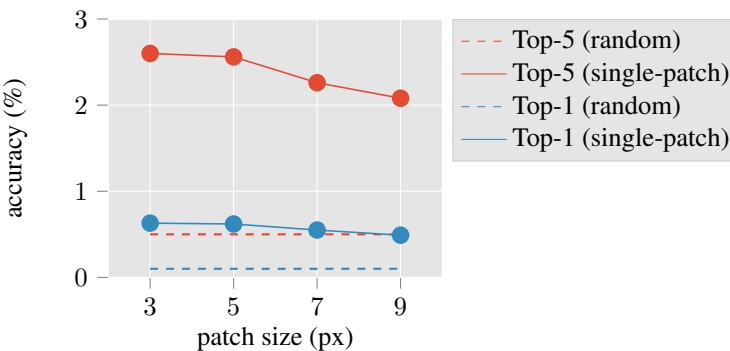

Figure 2: Training a ReLU neural network, with two hidden-layers of size 512, on a single patch of size $k \times k$ from the ImageNet data. The patch is randomly chosen from inside the image. We train the networks with Adam, with batch size of 50, for 10k iterations.

## B    PROOFS OF SECTION 4

We assume w.l.o.g that for every $i, j$ such that $\mathcal{I}_{i,j} = 0$, the $(i, j)$ gate is constant $\gamma_{i,j} \equiv \text{sign}(\mathbb{E}_{\mathcal{D}}[y])$. Since the output of this gate has no influence on the output $y$, we can choose it freely without changing the target function. To prove Theorem 1 and Theorem 2, we observe the behavior of the algorithm on the $i$-th layer. Let $\psi : \{\pm 1\}^{n_i} \to \{\pm 1\}^{n_i}$ be some mapping such that $\psi(\boldsymbol{x}) = (\xi_1 \cdot x_1, \dots, \xi_{n_i} \cdot x_{n_i})$ for $\xi_1, \dots, \xi_{n_i} \in \{\pm 1\}$. We also define $\varphi_i : \{\pm 1\}^{n_i/2} \to \{\pm 1\}^{n_i/2}$ such that:

$$\varphi_i(\boldsymbol{z}) = (\nu_1 z_1, \dots, \nu_{n_i/2} z_{n_i/2})$$

where $\nu_j := \begin{cases} \text{sign}(c_{i-1,j}) & c_{i-1,j} \neq 0 \\ 1 & \mathcal{I}_{i-1,j} = 0 \end{cases}$

Fix some $\epsilon' > 0$. We need to handle "bad" examples - examples that "rare" patches appear in them. For every $(i, j)$ gate, we observe all the input patterns $\boldsymbol{p}$ to the $(i, j)$ gate that appear with probability at most $\epsilon'$. Denote the following set of triplets:

$$\widetilde{\mathcal{P}}_{\epsilon'} := \left\{ (i, j, \boldsymbol{p}) \ : \ \mathbb{P}_{(\boldsymbol{x}, y) \sim \mathcal{D}^{(i)}} \left[ (x_{2j-1}, x_{2j}) = \boldsymbol{p} \right] < \epsilon' \right\}$$

Denote the following set of "bad" examples:

$$\widetilde{\mathcal{X}}_{\epsilon'} := \left\{ \boldsymbol{x} \in \mathcal{X} \ : \ \exists (i, j, \boldsymbol{p}) \in \widetilde{\mathcal{P}}_{\epsilon'} \ s.t. \ (z_{2j-1}, z_{2j}) = \boldsymbol{p} \ for \ \boldsymbol{z} = \Gamma_{(i+1)\dots d}(\boldsymbol{x}) \right\}$$

We have the following important result, which we prove in the sequel:

**Lemma 6.** *Fix $\epsilon > 0$ and let $\epsilon' \leq \epsilon$ such that $\mathbb{P}_{(\boldsymbol{x}, y) \sim \mathcal{D}} \left[ \boldsymbol{x} \in \widetilde{\mathcal{X}}_{\epsilon'} \right] < \frac{\epsilon}{8\sqrt{2}n_i} \min\{\Delta, 2\beta\}$. Assume we initialize $\boldsymbol{w}_l^{(0)}$ such that $\left\| \boldsymbol{w}_l^{(0)} \right\| \leq \frac{1}{4k}$. Fix $\delta > 0$. Assume we sample $S \sim \mathcal{D}$, with $|S| > \frac{128}{\epsilon^2 \min\{\Delta, 2\beta\}^2} \log(\frac{8n_i}{\delta})$. Assume that $k \geq \log^{-1}(\frac{4}{3}) \log(\frac{8n_i}{\delta})$, and that $\eta \leq \frac{n_i}{16k}$. Let $\Psi : \mathcal{X} \to [-1, 1]^{n_i/2}$ such that for every $\boldsymbol{x} \notin \widetilde{\mathcal{X}}_{\epsilon'}$ we have $\Psi(\boldsymbol{x}) = \psi \circ \Gamma_{(i+1)\dots d}(\boldsymbol{x})$ for some $\psi$ as defined above. Assume we perform the following updates:*

$$\boldsymbol{W}_t^{(i)} \leftarrow \boldsymbol{W}_{t-1}^{(i)} - \eta \frac{\partial}{\partial \boldsymbol{W}_{t-1}^{(i)}} L_S(P(B_{\boldsymbol{W}_{t-1}^{(i)}, \boldsymbol{V}_0^{(i)}}))$$

*Then with probability at least $1 - \delta$, for $t > \frac{3n_i}{\sqrt{2}\eta\epsilon \min\{\Delta, 2\beta\}}$ we have: $B_{\boldsymbol{W}_t^{(i)}, \boldsymbol{V}_0^{(i)}}(\boldsymbol{x}) = \varphi_i \circ \Gamma_i \circ \psi(\boldsymbol{x})$ for every $\boldsymbol{x} \notin \widetilde{\mathcal{X}}_{\epsilon}$.*

Given these results, we can prove the main theorems:

*Proof.* of Theorem 1 and Theorem 2. Fix $\delta' = \frac{\delta}{d}$. Let $\epsilon_0 \geq \cdots \geq \epsilon_d > 0$ such that for every $i \in [d]$ we have: $\mathbb{P}_{(\boldsymbol{x},y)\sim\mathcal{D}}\left[\boldsymbol{x} \in \widetilde{\mathcal{X}}_{\epsilon_i}\right] < \frac{\epsilon_{i-1}\min\{\Delta,2\beta\}}{8\sqrt{2}n_i}$ (we will note the exact value of $\epsilon_i$ later). We show that for every $i \in [d]$, w.p at least $1 - (d - i + 1)\delta'$, after the $i$-th step of the algorithm we have $\mathcal{N}_{i-1}(\boldsymbol{x}) = \varphi_i \circ \Gamma_{i\ldots d}(\boldsymbol{x})$ for every $\boldsymbol{x} \notin \widetilde{\mathcal{X}}_{\epsilon_{i-1}}$. By induction on $i$:

- For $i = d$, we get the required using Lemma 6 with $\psi, \Psi = id$ and $\epsilon = \epsilon_{d-1}$, $\epsilon' = \epsilon_d$.

- Assume the above holds for $i$, and we show it for $i - 1$. By the assumption, w.p at least $1 - (d - i + 1)\delta'$ we have $\mathcal{N}_{i-1}(\boldsymbol{x}) = \varphi_i \circ \Gamma_{i\ldots d}(\boldsymbol{x})$ for every $\boldsymbol{x} \notin \widetilde{\mathcal{X}}_{\epsilon_{i-1}}$. Observe that:

$$\frac{\partial L_{\mathcal{D}}}{\partial \boldsymbol{W}_t^{(i-1)}}(P(B_{\boldsymbol{W}_{t-1}^{(i-1)},\boldsymbol{V}_0^{(i-1)}} \circ \mathcal{N}_{i-1})) = \frac{\partial L_{\mathcal{N}_{i-1}(\mathcal{D})}}{\partial \boldsymbol{W}_t^{(i-1)}}(P(B_{\boldsymbol{W}_t^{(i-1)},\boldsymbol{V}_0^{(i-1)}}))$$

So using Lemma 6 with $\psi = \varphi_i$, $\Psi = \mathcal{N}_i$ and $\epsilon = \epsilon_{i-2}$, $\epsilon' = \epsilon_{i-1}$ we get that w.p at least $1 - \delta'$ we have $B_{\boldsymbol{W}_T^{(i-1)},\boldsymbol{V}_0^{(i-1)}}(\boldsymbol{x}) = \varphi_{i-1} \circ \Gamma_{i-1} \circ \varphi_i(\boldsymbol{x})$ for every $\boldsymbol{x} \notin \widetilde{\mathcal{X}}_{\epsilon_{i-2}}$. In this case, since $\varphi_i \circ \varphi_i = id$, we get that for every $\boldsymbol{x} \notin \widetilde{\mathcal{X}}_{\epsilon_{i-2}}$:

$$\mathcal{N}_{i-2}(\boldsymbol{x}) = B_{\boldsymbol{W}_T^{(i-1)},\boldsymbol{V}_0^{(i-1)}} \circ \mathcal{N}_{i-1}(\boldsymbol{x})$$
$$= (\varphi_{i-1} \circ \Gamma_{i-1} \circ \varphi_i) \circ (\varphi_i \circ \Gamma_{i\ldots d})(\boldsymbol{x}) = \varphi_{i-1} \circ \Gamma_{(i-1)\ldots d}(\boldsymbol{x})$$

  and using the union bound gives the required.

Notice that $\varphi_1 = id$: by definition of $\mathcal{D}^{(0)} = \Gamma_{1\ldots d}(\mathcal{D})$, for $(\boldsymbol{z}, y) \sim \mathcal{D}^{(0)}$ we have $\boldsymbol{z} = \Gamma_{1\ldots d}(\boldsymbol{x})$ and also $y = \Gamma_{1\ldots d}(\boldsymbol{x})$ for $(\boldsymbol{x}, y) \sim \mathcal{D}$. Therefore, we have $c_{0,1} = \mathbb{E}_{(x,y)\sim\mathcal{D}^{(0)}}[xy] = 1$, and therefore $\varphi_i(z) = \text{sign}(c_{0,1})z = z$. Now, choosing $i = 1$, the above result shows that with probability at least $1 - \delta$, the algorithm returns $\mathcal{N}_0$ such that $\mathcal{N}_0(\boldsymbol{x}) = \varphi_1 \circ \Gamma_1 \circ \cdots \circ \Gamma_d(\boldsymbol{x}) = h_C(\boldsymbol{x})$ for every $\boldsymbol{x} \notin \widetilde{\mathcal{X}}_{\epsilon_0}$.

To prove Theorem 2, it is enough to observe that when taking $\epsilon_0 = \cdots = \epsilon_d = \epsilon$, assumption 3 implies that $\widetilde{P}_\epsilon = \emptyset$ and therefore $\widetilde{\mathcal{X}}_\epsilon = \emptyset$, so the theorem follows. To prove Theorem 1, we take $\epsilon_0 = \epsilon$ and inductively define $\epsilon_i = \frac{\epsilon_{i-1}\min\{\Delta,2\beta\}}{32\sqrt{2}n^2}$. Notice that $\left|\widetilde{\mathcal{P}}_{\epsilon_i}\right| \leq \sum_{i\in[d]} \frac{n_i}{2}\left|\{\pm1\}^2\right| = 4\sum_{i=0}^{\log n} 2^{i-1} = 4n$. So, using the union bound we get:

$$\mathbb{P}_{(\boldsymbol{x},y)\sim\mathcal{D}}\left[\boldsymbol{x} \in \widetilde{\mathcal{X}}_{\epsilon_i}\right] = \mathbb{P}_{(\boldsymbol{x},y)\sim\mathcal{D}}\left[\cup_{(i,j,\boldsymbol{p})\in\widetilde{\mathcal{P}}_{\epsilon_i}} \Gamma_{(i+1)\ldots d}(\boldsymbol{x})_{(2j-1,2j)} = \boldsymbol{p}\right]$$
$$\leq \sum_{(i,j,\boldsymbol{p})\in\widetilde{\mathcal{P}}_{\epsilon_i}} \mathbb{P}_{(\boldsymbol{x},y)\sim\mathcal{D}}\left[\Gamma_{(i+1)\ldots d}(\boldsymbol{x})_{(2j-1,2j)} = \boldsymbol{p}\right]$$
$$\leq \sum_{(i,j,\boldsymbol{p})\in\widetilde{\mathcal{P}}_{\epsilon_i}} \mathbb{P}_{(\boldsymbol{x},y)\sim\mathcal{D}^{(i)}}\left[(x_{2j-1}, x_{2j}) = \boldsymbol{p}\right] < \left|\widetilde{\mathcal{P}}_{\epsilon'}\right| \cdot \epsilon_i \leq \frac{\epsilon_{i-1}\Delta}{8\sqrt{2}n_i}$$

Now, observing that $\epsilon_d = \frac{\min\{\Delta,2\beta\}^d\epsilon}{2^{5.5d}n^{2d}} = \frac{n^{\log\min\{\Delta,2\beta\}}\epsilon}{n^{5.5+2\log n}}$ gives the required. $\qquad\square$

In the rest of this section we prove Lemma 6. Fix some $i \in [d]$ and let $j \in [n_i/2]$. With slight abuse of notation, we denote by $\boldsymbol{w}^{(t)}$ the value of the weight $\boldsymbol{w}^{(i,j)}$ at iteration $t$, and denote $\boldsymbol{v} := \boldsymbol{v}^{(i,j)}$ and $g_t := g_{\boldsymbol{w}^{(t)},\boldsymbol{v}}$. Recall that we defined $\psi(\boldsymbol{x}) = (\xi_1 \cdot x_1, \ldots, \xi_{n_i} \cdot x_{n_i})$ for $\xi_1 \ldots \xi_{n_i} \in \{\pm1\}$. Denote $\widetilde{\mathcal{D}}^{(i)} := \psi(\mathcal{D}^{(i)})$ the distribution of $(\psi(\boldsymbol{x}), y)$, where $(\boldsymbol{x}, y) \sim \mathcal{D}^{(i)}$. Let $\gamma := \gamma_{i-1,j}$, and let $\widetilde{\gamma}$ such that $\widetilde{\gamma}(x_1, x_2) = \gamma(\xi_{2j-1} \cdot x_1, \xi_{2j} \cdot x_2)$. For every $\boldsymbol{p} \in \{\pm1\}^2$, denote $\widetilde{\boldsymbol{p}} := (\xi_{2j-1}p_1, \xi_{2j}p_2)$, so we have $\gamma(\widetilde{\boldsymbol{p}}) = \widetilde{\gamma}(\boldsymbol{p})$. Then we have the following:

**Lemma 7.** *Fix some $\boldsymbol{p} \in \{\pm1\}^2$ such that $(i, j, \widetilde{\boldsymbol{p}}) \notin \widetilde{\mathcal{P}}_\epsilon$. For every $l \in [k]$ such that $\langle \boldsymbol{w}_l^{(t)}, \boldsymbol{p}\rangle > 0$ and $g_t(\boldsymbol{p}) \in (-1, 1)$, the following holds:*

$$-\widetilde{\gamma}(\boldsymbol{p})v_l\nu_j\langle\frac{\partial L_{\widetilde{\mathcal{D}}^{(i)}}}{\partial\boldsymbol{w}_l^{(t)}}, \boldsymbol{p}\rangle > \frac{\sqrt{2}\epsilon}{n_i}\min\{\Delta, 2\beta\}$$

*Proof.* Observe the following:

$$\frac{\partial L_{\widetilde{\mathcal{D}}^{(i)}}}{\partial \boldsymbol{w}_l^{(t)}}(P(B_{\boldsymbol{W}^{(i)}, \boldsymbol{V}^{(i)}})) = \mathbb{E}_{(\boldsymbol{x}, y) \sim \widetilde{\mathcal{D}}^{(i)}} \left[ \ell'(P(B_{\boldsymbol{W}^{(i)}, \boldsymbol{V}^{(i)}})(\boldsymbol{x})) \cdot \frac{\partial}{\partial \boldsymbol{w}_l^{(t)}} \frac{2}{n_i} \sum_{j'=1}^{n_i/2} g_{\boldsymbol{w}^{(i,j')}, \boldsymbol{v}^{(i,j')}}(x_{2j'-1}, x_{2j'}) \right]$$

$$= \frac{2}{n_i} \mathbb{E}_{(\boldsymbol{x}, y) \sim \widetilde{\mathcal{D}}^{(i)}} \left[ -y \frac{\partial}{\partial \boldsymbol{w}_l^{(t)}} g_t(x_{2j-1}, x_{2j}) \right]$$

$$= \frac{2}{n_i} \mathbb{E}_{(\boldsymbol{x}, y) \sim \widetilde{\mathcal{D}}^{(i)}} \left[ -y v_l \mathbf{1} \{ g_t(x_{2j-1}, x_{2j}) \in (-1, 1) \} \cdot \mathbf{1} \{ \langle \boldsymbol{w}_l^{(t)}, (x_{2j-1}, x_{2j}) \rangle > 0 \} \cdot (x_{2j-1}, x_{2j}) \right]$$

We use the fact that $\ell'(P(B_{\boldsymbol{W}^{(i)}, \boldsymbol{V}^{(i)}})(\boldsymbol{x})) = -y$, unless $P(B_{\boldsymbol{W}^{(i)}, \boldsymbol{V}^{(i)}})(\boldsymbol{x}) \in \{\pm 1\}$, in which case $g_t(x_{2j-1}, x_{2j}) \in \{\pm 1\}$, so $\frac{\partial}{\partial \boldsymbol{w}_l^{(t)}} g_t(x_{2j-1}, x_{2j}) = 0$. Fix some $\boldsymbol{p} \in \{\pm 1\}^2$ such that $\langle \boldsymbol{w}_l^{(t)}, \boldsymbol{p} \rangle > 0$. Note that for every $\boldsymbol{p} \neq \boldsymbol{p}' \in \{\pm 1\}^2$ we have either $\langle \boldsymbol{p}, \boldsymbol{p}' \rangle = 0$, or $\boldsymbol{p} = -\boldsymbol{p}'$ in which case $\langle \boldsymbol{w}_l^{(t)}, \boldsymbol{p}' \rangle < 0$. Therefore, we get the following:

$$\langle \frac{\partial L_{\widetilde{\mathcal{D}}^{(i)}}}{\partial \boldsymbol{w}_l^{(t)}}, \boldsymbol{p} \rangle = \frac{2}{n_i} \mathbb{E}_{(\boldsymbol{x}, y) \sim \widetilde{\mathcal{D}}^{(i)}} \left[ -y v_l \mathbf{1} \{ g_t(x_{2j-1}, x_{2j}) \in (-1, 1) \} \cdot \mathbf{1} \{ \langle \boldsymbol{w}_l^{(t)}, (x_{2j-1}, x_{2j}) \rangle \geq 0 \} \cdot \langle (x_{2j-1}, x_{2j}), \boldsymbol{p} \rangle \right]$$

$$= \frac{2}{n_i} \mathbb{E}_{(\boldsymbol{x}, y) \sim \widetilde{\mathcal{D}}^{(i)}} \left[ -y v_l \mathbf{1} \{ g_t(x_{2j-1}, x_{2j}) \in (-1, 1) \} \cdot \mathbf{1} \{ (x_{2j-1}, x_{2j}) = \boldsymbol{p} \} \|\boldsymbol{p}\| \right]$$

Denote $q_{\boldsymbol{p}} := \mathbb{P}_{(\boldsymbol{x}, y) \sim \mathcal{D}^{(i)}} [(x_{2j-1}, x_{2j}) = \boldsymbol{p} | \gamma(x_{2j-1}, x_{2j}) = \gamma(\boldsymbol{p})]$. Using property 2, we have:

$$\mathbb{P}_{(\boldsymbol{x}, y) \sim \mathcal{D}^{(i)}} [(x_{2j-1}, x_{2j}) = \boldsymbol{p}, y = y']$$
$$= \mathbb{P}_{(\boldsymbol{x}, y) \sim \mathcal{D}^{(i)}} [(x_{2j-1}, x_{2j}) = \boldsymbol{p}, y = y', \gamma(x_{2j-1}, x_{2j}) = \gamma(\boldsymbol{p})]$$
$$= \mathbb{P}_{(\boldsymbol{x}, y) \sim \mathcal{D}^{(i)}} [(x_{2j-1}, x_{2j}) = \boldsymbol{p}, y = y' | \gamma(x_{2j-1}, x_{2j}) = \gamma(\boldsymbol{p})] \mathbb{P}_{(\boldsymbol{x}, y) \sim \mathcal{D}^{(i)}} [\gamma(x_{2j-1}, x_{2j}) = \gamma(\boldsymbol{p})]$$
$$= q_{\boldsymbol{p}} \mathbb{P}_{(\boldsymbol{x}, y) \sim \mathcal{D}^{(i)}} [\gamma(x_{2j-1}, x_{2j}) = \gamma(\boldsymbol{p}), y = y']$$
$$= q_{\boldsymbol{p}} \mathbb{P}_{(\boldsymbol{z}, y) \sim \mathcal{D}^{(i-1)}} [z_j = \gamma(\boldsymbol{p}), y = y']$$

And therefore:

$$\mathbb{E}_{(\boldsymbol{x}, y) \sim \mathcal{D}^{(i)}} [y \mathbf{1} \{ (x_{2j-1}, x_{2j}) = \boldsymbol{p} \}] = \sum_{y' \in \{\pm 1\}} y' \mathbb{P}_{(\boldsymbol{x}, y) \sim \mathcal{D}^{(i)}} [(x_{2j-1}, x_{2j}) = \boldsymbol{p}, y = y']$$

$$= q_{\boldsymbol{p}} \sum_{y' \in \{\pm 1\}} y' \mathbb{P}_{(\boldsymbol{z}, y) \sim \mathcal{D}^{(i-1)}} [z_j = \gamma(\boldsymbol{p}), y = y']$$

$$= q_{\boldsymbol{p}} \mathbb{E}_{(\boldsymbol{z}, y) \sim \mathcal{D}^{(i-1)}} [y \mathbf{1} \{ z_j = \gamma(\boldsymbol{p}) \}]$$

Assuming $g_t(\boldsymbol{p}) \in (-1, 1)$, using the above we get:

$$\langle \frac{\partial L_{\widetilde{\mathcal{D}}^{(i)}}}{\partial \boldsymbol{w}_l^{(t)}}, \boldsymbol{p} \rangle = \frac{2\sqrt{2} v_l}{n_i} \mathbb{E}_{(\boldsymbol{x}, y) \sim \widetilde{\mathcal{D}}^{(i)}} [-y \mathbf{1} \{ (x_{2j-1}, x_{2j}) = \boldsymbol{p} \}]$$

$$= \frac{2\sqrt{2} v_l}{n_i} \mathbb{E}_{(\boldsymbol{x}, y) \sim \mathcal{D}^{(i)}} [-y \mathbf{1} \{ (\xi_{2j-1} x_{2j-1}, \xi_{2j} x_{2j}) = \boldsymbol{p} \}]$$

$$= -\frac{2\sqrt{2} v_l}{n_i} \mathbb{E}_{(\boldsymbol{x}, y) \sim \mathcal{D}^{(i)}} [y \mathbf{1} \{ (x_{2j-1}, x_{2j}) = \widehat{\boldsymbol{p}} \}]$$

$$= -\frac{2\sqrt{2} v_l q_{\widehat{\boldsymbol{p}}}}{n_i} \mathbb{E}_{(\boldsymbol{z}, y) \sim \mathcal{D}^{(i-1)}} [y \mathbf{1} \{ z_j = \widetilde{\gamma}(\boldsymbol{p}) \}]$$

Now, we have the following cases:

- If $\mathcal{I}_{i-1,j} = 0$, then by property 1 $z_j$ and $y$ are independent, so:

$$\langle \frac{\partial L_{\widetilde{\mathcal{D}}^{(i)}}}{\partial \boldsymbol{w}_l^{(t)}}, \boldsymbol{p} \rangle = -\frac{2\sqrt{2}v_l q_{\widetilde{\boldsymbol{p}}}}{n_i} \mathbb{E}_{(\boldsymbol{z},y)\sim\mathcal{D}^{(i-1)}} [y\mathbf{1}\{z_j = \widetilde{\gamma}(\boldsymbol{p})\}]$$

$$= -\frac{2\sqrt{2}v_l q_{\widetilde{\boldsymbol{p}}}}{n_i} \mathbb{E}_{(\boldsymbol{z},y)\sim\mathcal{D}^{(i-1)}} [y] \, \mathbb{P}_{(\boldsymbol{z},y)\sim\mathcal{D}^{(i-1)}} [z_j = \widetilde{\gamma}(\boldsymbol{p})]$$

$$= -\frac{2\sqrt{2}v_l}{n_i} \mathbb{E}_{(\boldsymbol{z},y)\sim\mathcal{D}^{(i-1)}} [y] \, \mathbb{P}_{(\boldsymbol{z},y)\sim\mathcal{D}^{(i-1)}} [(x_{2j-1}, x_{2j}) = \widetilde{\boldsymbol{p}}]$$

Since we assume $\widetilde{\gamma}(\boldsymbol{p}) = \text{sign}(\mathbb{E}[y])$, and using the fact that $(i, j, \widetilde{\boldsymbol{p}}) \notin \widetilde{\mathcal{P}}_\epsilon$, we get that:

$$-\widetilde{\gamma}(\boldsymbol{p})v_l\nu_j\langle \frac{\partial L_{\widetilde{\mathcal{D}}^{(i)}}}{\partial \boldsymbol{w}_l^{(t)}}, \boldsymbol{p} \rangle = -\text{sign}(\mathbb{E}[y])v_l\langle \frac{\partial L_{\widetilde{\mathcal{D}}^{(i)}}}{\partial \boldsymbol{w}_l^{(t)}}, \boldsymbol{p} \rangle$$

$$= \frac{2\sqrt{2}}{n_i} |\mathbb{E}[y]| \, \mathbb{P}_{(\boldsymbol{z},y)\sim\mathcal{D}^{(i-1)}} [(x_{2j-1}, x_{2j}) = \widetilde{\boldsymbol{p}}] > \frac{2\sqrt{2}}{n_i}\beta\epsilon$$

- Otherwise, observe that:

$$\langle \frac{\partial L_{\widetilde{\mathcal{D}}^{(i)}}}{\partial \boldsymbol{w}_l^{(t)}}, \boldsymbol{p} \rangle = -\frac{2\sqrt{2}v_l q_{\widetilde{\boldsymbol{p}}}}{n_i} \mathbb{E}_{(\boldsymbol{z},y)\sim\mathcal{D}^{(i-1)}} [y\mathbf{1}\{z_j = \widetilde{\gamma}(\boldsymbol{p})\}]$$

$$= -\frac{2\sqrt{2}v_l q_{\widetilde{\boldsymbol{p}}}}{n_i} \mathbb{E}_{(\boldsymbol{z},y)\sim\mathcal{D}^{(i-1)}} \left[ y\frac{1}{2}(z_j \cdot \widetilde{\gamma}(\boldsymbol{p}) + 1) \right]$$

$$= -\frac{\sqrt{2}v_l q_{\widetilde{\boldsymbol{p}}}}{n_i} \left( \widetilde{\gamma}(\boldsymbol{p})c_{i-1,j} + \mathbb{E}_{(\boldsymbol{z},y)\sim\mathcal{D}^{(i-1)}} [y] \right)$$

And therefore, using property 1, since $\mathcal{I}_{i-1,j} \neq 0$, we get:

$$-\widetilde{\gamma}(\boldsymbol{p})v_l \text{sign}(c_{i-1,j})\langle \frac{\partial L_{\widetilde{\mathcal{D}}^{(i)}}}{\partial \boldsymbol{w}_l^{(t)}}, \boldsymbol{p} \rangle = \frac{\sqrt{2}q_{\widetilde{\boldsymbol{p}}}}{n_i} \left( |c_{i-1,j}| + \text{sign}(c_{i-1,j})\widetilde{\gamma}(\boldsymbol{p})\mathbb{E}[y] \right)$$

$$\geq \frac{\sqrt{2}q_{\widetilde{\boldsymbol{p}}}}{n_i} \left( |c_{i-1,j}| - |\mathbb{E}[y]| \right) > \frac{\sqrt{2}\epsilon}{n_i}\Delta$$

where we use the fact that $(i, j, \widetilde{\boldsymbol{p}}) \notin \widetilde{\mathcal{P}}_\epsilon$.

$\square$

We introduce the following notation: for a sample $S \subseteq \mathcal{X}' \times \mathcal{Y}$, and some function $f : \mathcal{X}' \to \mathcal{X}'$, denote by $f(S)$ the sample $f(S) := \{(f(\boldsymbol{x}), y)\}_{(\boldsymbol{x},y)\in S}$.

**Lemma 8.** *Fix $\delta > 0$. Assume we sample $S \sim \mathcal{D}$, with $|S| > \frac{128}{\epsilon^2 \min\{\Delta, 2\beta\}^2} \log \frac{4}{\delta}$. Then, with probability at least $1 - \delta$, for every $\boldsymbol{p} \in \{\pm 1\}^2$ such that $\langle \boldsymbol{w}_l^{(t)}, \boldsymbol{p} \rangle > 0$ it holds that:*

$$\left| \langle \frac{\partial L_{\Psi(\mathcal{D})}}{\partial \boldsymbol{w}_l^{(t)}}, \boldsymbol{p} \rangle - \langle \frac{\partial L_{\Psi(S)}}{\partial \boldsymbol{w}_l^{(t)}}, \boldsymbol{p} \rangle \right| \leq \frac{\epsilon}{2\sqrt{2}n_i} \min\{\Delta, 2\beta\}$$

*Proof.* Fix some $\boldsymbol{p} \in \{\pm 1\}^2$ with $\langle \boldsymbol{w}_l^{(t)}, \boldsymbol{p} \rangle > 0$. Similar to what we previously showed, we get that:

$$\langle \frac{\partial L_{\Psi(S)}}{\partial \boldsymbol{w}_l^{(t)}}, \boldsymbol{p} \rangle = \frac{2}{n_i} \mathbb{E}_{(\boldsymbol{x},y) \sim \Psi(S)} \left[ -yv_l \mathbf{1}\{g_t(x_{2j-1}, x_{2j}) \in (-1,1)\} \cdot \mathbf{1}\{\langle \boldsymbol{w}_l^{(t)}, (x_{2j-1}, x_{2j}) \rangle \geq 0\} \cdot \langle (x_{2j-1}, x_{2j}), \boldsymbol{p} \rangle \right]$$

$$= \frac{2}{n_i} \mathbb{E}_{(\boldsymbol{x},y) \sim \Psi(S)} \left[ -yv_l \mathbf{1}\{g_t(x_{2j-1}, x_{2j}) \in (-1,1)\} \cdot \mathbf{1}\{(x_{2j-1}, x_{2j}) = \boldsymbol{p}\} \|\boldsymbol{p}\| \right]$$

$$= \frac{2\sqrt{2}}{n_i} \mathbb{E}_{(\boldsymbol{x},y) \sim \Psi(S)} \left[ -yv_l \mathbf{1}\{g_t(x_{2j-1}, x_{2j}) \in (-1,1)\} \cdot \mathbf{1}\{(x_{2j-1}, x_{2j}) = \boldsymbol{p}\} \right]$$

Denote $f(\boldsymbol{x}, y) = -yv_l \mathbf{1}\{g_t(x_{2j-1}, x_{2j}) \in (-1,1)\} \cdot \mathbf{1}\{(x_{2j-1}, x_{2j}) = \boldsymbol{p}\}$, and notice that $f(\boldsymbol{x}, y) \in [-1, 1]$. Now, from Hoeffding's inequality we get that:

$$\mathbb{P}_S \left[ |\mathbb{E}_{\Psi(S)}[f(\boldsymbol{x}, y)] - \mathbb{E}_{\Psi(\mathcal{D})}[f(\boldsymbol{x}, y)]| \geq \tau \right] \leq \exp\left( -\frac{1}{2}|S|\tau^2 \right)$$

So, for $|S| > \frac{2}{\tau^2} \log \frac{4}{\delta}$ we get that with probability at least $1 - \frac{\delta}{4}$ we have:

$$\left| \langle \frac{\partial L_{\Psi(\mathcal{D})}}{\partial \boldsymbol{w}_l^{(t)}}, \boldsymbol{p} \rangle - \langle \frac{\partial L_{\Psi(S)}}{\partial \boldsymbol{w}_l^{(t)}}, \boldsymbol{p} \rangle \right| = \frac{2\sqrt{2}}{n_i} \left| \mathbb{E}_{\Psi(S)}[f(\boldsymbol{x}, y)] - \mathbb{E}_{\Psi(\mathcal{D})}[f(\boldsymbol{x}, y)] \right| < \frac{2\sqrt{2}}{n_i} \tau$$

Taking $\tau = \frac{\epsilon}{8} \min\{\Delta, 2\beta\}$ and using the union bound over all $\boldsymbol{p} \in \{\pm 1\}^2$ completes the proof. $\square$

**Lemma 9.** *Fix $\delta > 0$. Assume $\mathbb{P}_{(\boldsymbol{x},y) \sim \mathcal{D}}\left[ \boldsymbol{x} \in \widetilde{\mathcal{X}}_{\epsilon'} \right] < \frac{\epsilon}{8\sqrt{2}n_i} \min\{\Delta, 2\beta\}$. Assume we sample $S \sim \mathcal{D}$, with $|S| > \frac{128}{\epsilon^2 \min\{\Delta, 2\beta\}^2} \log \frac{4}{\delta}$. Then, with probability at least $1 - \delta$, for every $\boldsymbol{p} \in \{\pm 1\}^2$ such that $(i, j, \widetilde{\boldsymbol{p}}) \notin \widetilde{\mathcal{P}}_{\epsilon'}$, and for every $l \in [k]$ such that $\langle \boldsymbol{w}_l^{(t)}, \boldsymbol{p} \rangle > 0$ and $g_t(\boldsymbol{p}) \in (-1, 1)$, the following holds:*

$$-\widetilde{\gamma}(\boldsymbol{p})v_l \nu_j \langle \frac{\partial L_{\Psi(\mathcal{D})}}{\partial \boldsymbol{w}_l^{(t)}}, \boldsymbol{p} \rangle > \frac{\epsilon}{\sqrt{2}n_i} \min\{\Delta, 2\beta\}$$

*Proof.* Denote $\alpha := \mathbb{P}_{(\boldsymbol{x},y) \sim \mathcal{D}}\left[ \boldsymbol{x} \in \widetilde{\mathcal{X}}_{\epsilon'} \right]$, and denote $B^{(i)} := B_{\boldsymbol{W}^{(i)}, \boldsymbol{V}^{(i)}}$. Then, we have the following:

$$\left\| \frac{\partial L_{\widetilde{\mathcal{D}}^{(i)}}}{\partial \boldsymbol{w}_l^{(t)}} - \frac{\partial L_{\Psi(\mathcal{D})}}{\partial \boldsymbol{w}_l^{(t)}} \right\| = \left\| \mathbb{E}_{(\boldsymbol{x},y) \sim \mathcal{D}} \left[ \frac{\partial}{\partial \boldsymbol{w}_l^{(t)}} \ell(PB^{(i)} \circ \psi \circ \Gamma_{(i+1)...d}(\boldsymbol{x})) - \frac{\partial}{\partial \boldsymbol{w}_l^{(t)}} \ell(PB^{(i)} \circ \Psi(\boldsymbol{x})) \right] \right\|$$

$$\leq \mathbb{E}_{(\boldsymbol{x},y) \sim \mathcal{D}} \left[ \left\| \frac{\partial}{\partial \boldsymbol{w}_l^{(t)}} \ell(PB^{(i)} \circ \psi \circ \Gamma_{(i+1)...d}(\boldsymbol{x})) - \frac{\partial}{\partial \boldsymbol{w}_l^{(t)}} \ell(PB^{(i)} \circ \Psi(\boldsymbol{x})) \right\| \cdot \mathbf{1}\left\{ \boldsymbol{x} \in \widetilde{\mathcal{X}}_{\epsilon'} \right\} \right]$$

$$\leq \mathbb{E}_{(\boldsymbol{x},y) \sim \mathcal{D}} \left[ \left\| \psi \circ \Gamma_{(i+1)...d}(\boldsymbol{x})_{(2j-1, 2j)} - \Psi(\boldsymbol{x})_{(2j-1, 2j)} \right\| \cdot \mathbf{1}\left\{ \boldsymbol{x} \in \widetilde{\mathcal{X}}_{\epsilon'} \right\} \right]$$

$$\leq 2\sqrt{2} \mathbb{P}_{(\boldsymbol{x},y) \sim \mathcal{D}}\left[ \boldsymbol{x} \in \widetilde{\mathcal{X}}_{\epsilon'} \right] = 2\sqrt{2}\alpha$$

So we get, using Lemma 7 and Lemma 8, with probability at least $1 - \delta$:

$$-\widetilde{\gamma}(\boldsymbol{p})v_l \nu_j \langle \frac{\partial L_{\Psi(S)}}{\partial \boldsymbol{w}_l^{(t)}}, \boldsymbol{p} \rangle = -\widetilde{\gamma}(\boldsymbol{p})v_l \nu_j \left( \langle \frac{\partial L_{\widetilde{\mathcal{D}}^{(i)}}}{\partial \boldsymbol{w}_l^{(t)}}, \boldsymbol{p} \rangle + \langle \frac{\partial L_{\Psi(S)}}{\partial \boldsymbol{w}_l^{(t)}}, \boldsymbol{p} \rangle - \langle \frac{\partial L_{\Psi(\mathcal{D})}}{\partial \boldsymbol{w}_l^{(t)}}, \boldsymbol{p} \rangle + \langle \frac{\partial L_{\Psi(\mathcal{D})}}{\partial \boldsymbol{w}_l^{(t)}}, \boldsymbol{p} \rangle - \langle \frac{\partial L_{\widetilde{\mathcal{D}}^{(i)}}}{\partial \boldsymbol{w}_l^{(t)}}, \boldsymbol{p} \rangle \right)$$

$$\geq -\widetilde{\gamma}(\boldsymbol{p})v_l \nu_j \langle \frac{\partial L_{\widetilde{\mathcal{D}}^{(i)}}}{\partial \boldsymbol{w}_l^{(t)}}, \boldsymbol{p} \rangle - \left| \langle \frac{\partial L_{\Psi(S)}}{\partial \boldsymbol{w}_l^{(t)}}, \boldsymbol{p} \rangle - \langle \frac{\partial L_{\Psi(\mathcal{D})}}{\partial \boldsymbol{w}_l^{(t)}}, \boldsymbol{p} \rangle \right| - \left| \langle \frac{\partial L_{\Psi(\mathcal{D})}}{\partial \boldsymbol{w}_l^{(t)}}, \boldsymbol{p} \rangle - \langle \frac{\partial L_{\widetilde{\mathcal{D}}^{(i)}}}{\partial \boldsymbol{w}_l^{(t)}}, \boldsymbol{p} \rangle \right|$$

$$> \frac{\sqrt{2}\epsilon}{n_i} \min\{\Delta, 2\beta\} - \frac{\epsilon}{2\sqrt{2}n_i} \min\{\Delta, 2\beta\} - \left\| \frac{\partial L_{\widetilde{\mathcal{D}}^{(i)}}}{\partial \boldsymbol{w}_l^{(t)}} - \frac{\partial L_{\Psi(\mathcal{D})}}{\partial \boldsymbol{w}_l^{(t)}} \right\| \|\boldsymbol{p}\|$$

$$\geq \frac{3\epsilon}{2\sqrt{2}n_i} \min\{\Delta, 2\beta\} - 4\alpha$$

So for $\alpha < \frac{\epsilon}{8\sqrt{2}n_i} \min\{\Delta, 2\beta\}$ we get the required. $\square$

We want to show that if the value of $g_t$ gets "stuck", then it recovered the value of the gate, multiplied by the correlation $c_{i-1,j}$. We do this by observing the dynamics of $\langle \boldsymbol{w}_l^{(t)}, \boldsymbol{p} \rangle$. In most cases, its value moves in the right direction, except for a small set that oscillates around zero. This set is the following:

$$A_t = \left\{ (l, \boldsymbol{p}) \; : \; (i, j, \widetilde{\boldsymbol{p}}) \notin \widetilde{\mathcal{P}}_\epsilon \wedge \widetilde{\gamma}(\boldsymbol{p}) v_l \nu_j < 0 \wedge \langle \boldsymbol{w}_l^{(t)}, \boldsymbol{p} \rangle \leq \frac{4\eta}{n_i} \wedge \left( \widetilde{\gamma}(-\boldsymbol{p}) v_l \nu_j < 0 \vee (i, j, -\widetilde{\boldsymbol{p}}) \in \widetilde{\mathcal{P}}_\epsilon \right) \right\}$$

We have the following simple observation:

**Lemma 10.** *With the assumptions of Lemma 9, with probability at least $1 - \delta$, for every $t$ we have:* $A_t \subseteq A_{t+1}$.

*Proof.* Fix some $(l, \boldsymbol{p}) \in A_t$, and we need to show that $\langle \boldsymbol{w}_l^{(t+1)}, \boldsymbol{p} \rangle \leq \frac{4\eta}{n_i}$. If $\langle \boldsymbol{w}_l^{(t)}, \boldsymbol{p} \rangle = 0$ then[4] $\langle \boldsymbol{w}_l^{(t+1)}, \boldsymbol{p} \rangle = \langle \boldsymbol{w}_l^{(t)}, \boldsymbol{p} \rangle \leq \frac{4\eta}{n_i}$ and we are done. If $\langle \boldsymbol{w}_l^{(t)}, \boldsymbol{p} \rangle > 0$ then, since $(i, j, \widetilde{\boldsymbol{p}}) \notin \widetilde{\mathcal{P}}_\epsilon$ we have from Lemma 9, w.p at least $1 - \delta$:

$$-\langle \frac{\partial L_{\Psi(S)}}{\partial \boldsymbol{w}_l^{(t)}}, \boldsymbol{p} \rangle < \widetilde{\gamma}(\boldsymbol{p}) v_l \nu_j \frac{\epsilon}{\sqrt{2} n_i} \min\{\Delta, 2\beta\} < 0$$

Where we use the fact that $\widetilde{\gamma}(\boldsymbol{p}) v_l \nu_j < 0$. Therefore, we get:

$$\langle \boldsymbol{w}_l^{(t+1)}, \boldsymbol{p} \rangle = \langle \boldsymbol{w}_l^{(t)}, \boldsymbol{p} \rangle - \eta \langle \frac{\partial L_{\Psi(S)}}{\partial \boldsymbol{w}_l^{(t)}}, \boldsymbol{p} \rangle \leq \langle \boldsymbol{w}_l^{(t)}, \boldsymbol{p} \rangle \leq \frac{4\eta}{n_i}$$

Otherwise, we have $\langle \boldsymbol{w}_l^{(t)}, \boldsymbol{p} \rangle < 0$, so:

$$\langle \boldsymbol{w}_l^{(t+1)}, \boldsymbol{p} \rangle = \langle \boldsymbol{w}_l^{(t)}, \boldsymbol{p} \rangle - \eta \langle \frac{\partial L_{\Psi(S)}}{\partial \boldsymbol{w}_l^{(t)}}, \boldsymbol{p} \rangle \leq \langle \boldsymbol{w}_l^{(t)}, \boldsymbol{p} \rangle + \frac{4\eta}{n_i} \leq \frac{4\eta}{n_i}$$

$\square$

Now, we want to show that all $\langle \boldsymbol{w}_l^{(t)}, \boldsymbol{p} \rangle$ with $(l, \boldsymbol{p}) \notin A_t$ and $(i, j, \widetilde{\boldsymbol{p}}) \notin \widetilde{\mathcal{P}}_\epsilon$ move in the direction of $\widetilde{\gamma}(\boldsymbol{p}) \cdot \nu_j$:

**Lemma 11.** *With the assumptions of Lemma 9, with probability at least $1 - \delta$, for every $l, t$ and $\boldsymbol{p} \in \{\pm 1\}^2$ such that $\langle \boldsymbol{w}_l^{(t)}, \boldsymbol{p} \rangle > 0$, $(i, j, \widetilde{\boldsymbol{p}}) \notin \widetilde{\mathcal{P}}_\epsilon$ and $(l, \boldsymbol{p}) \notin A_t$, it holds that:*

$$\left( \sigma(\langle \boldsymbol{w}_l^{(t)}, \boldsymbol{p} \rangle) - \sigma(\langle \boldsymbol{w}_l^{(t-1)}, \boldsymbol{p} \rangle) \right) \cdot \widetilde{\gamma}(\boldsymbol{p}) v_l \nu_j \geq 0$$

*Proof.* Assume the result of Lemma 9 holds (this happens with probability at least $1 - \delta$). We cannot have $\langle \boldsymbol{w}_l^{(t-1)}, \boldsymbol{p} \rangle = 0$, since otherwise we would have $\langle \boldsymbol{w}_l^{(t)}, \boldsymbol{p} \rangle = 0$, contradicting the assumption. If $\langle \boldsymbol{w}_l^{(t-1)}, \boldsymbol{p} \rangle > 0$, since we require $\langle \boldsymbol{w}_l^{(t)}, \boldsymbol{p} \rangle > 0$ we get that:

$$\sigma(\langle \boldsymbol{w}_l^{(t)}, \boldsymbol{p} \rangle) - \sigma(\langle \boldsymbol{w}_l^{(t-1)}, \boldsymbol{p} \rangle) = \langle \boldsymbol{w}_l^{(t)} - \boldsymbol{w}_l^{(t-1)}, \boldsymbol{p} \rangle = -\eta \langle \frac{\partial L_{\Psi(S)}}{\partial \boldsymbol{w}_l^{(t-1)}}, \boldsymbol{p} \rangle$$

and the required follows from Lemma 9. Otherwise, we have $\langle \boldsymbol{w}_l^{(t-1)}, \boldsymbol{p} \rangle < 0$. We observe the following cases:

- If $\widetilde{\gamma}(\boldsymbol{p}) v_l \nu_j \geq 0$ then we are done, since:

$$\left( \sigma(\langle \boldsymbol{w}_l^{(t)}, \boldsymbol{p} \rangle) - \sigma(\langle \boldsymbol{w}_l^{(t-1)}, \boldsymbol{p} \rangle) \right) \cdot \widetilde{\gamma}(\boldsymbol{p}) \nu_j = \sigma(\langle \boldsymbol{w}_l^{(t)}, \boldsymbol{p} \rangle) \cdot \widetilde{\gamma}(\boldsymbol{p}) v_l \nu_j \geq 0$$

---

[4]Formally, there is no gradient, but we'll just take the sub-gradient zero.

- Otherwise, we have $\widetilde{\gamma}(\boldsymbol{p})v_l\nu_j < 0$. We also have:

$$\langle \boldsymbol{w}_l^{(t)}, \boldsymbol{p}\rangle = \langle \boldsymbol{w}_l^{(t-1)}, \boldsymbol{p}\rangle - \eta\langle\frac{\partial L_{\Psi(S)}}{\partial \boldsymbol{w}_l^{(t)}}, \boldsymbol{p}\rangle \le \langle \boldsymbol{w}_l^{(t-1)}, \boldsymbol{p}\rangle + \frac{4\eta}{n_i} \le \frac{4\eta}{n_i}$$

Since we assume $(l, \boldsymbol{p}) \notin A_t$, we must have $(i, j, -\widetilde{\boldsymbol{p}}) \notin \widetilde{\mathcal{P}}_\epsilon$ and $\widetilde{\gamma}(-\boldsymbol{p})v_l\nu_j \ge 0$. Therefore, from Lemma 9 we get:

$$\langle\frac{\partial L_{\Psi(S)}}{\partial \boldsymbol{w}_l^{(t)}}, -\boldsymbol{p}\rangle < -\widetilde{\gamma}(-\boldsymbol{p})v_l\nu_j\frac{\epsilon}{\sqrt{2}n_i}\min\{\Delta, 2\beta\}$$

And hence:

$$0 < \langle \boldsymbol{w}_l^{(t)}, \boldsymbol{p}\rangle = \langle \boldsymbol{w}_l^{(t-1)}, \boldsymbol{p}\rangle + \eta\langle\frac{\partial L_{\Psi(S)}}{\partial \boldsymbol{w}_l^{(t-1)}}, -\boldsymbol{p}\rangle \le -\eta\widetilde{\gamma}(-\boldsymbol{p})v_l\nu_j\frac{\epsilon}{\sqrt{2}n_i}\min\{\Delta, 2\beta\} < 0$$

and we reach a contradiction.

$\square$

From the above, we get the following:

**Corollary 3.** *With the assumptions of Lemma 9, with probability at least $1 - \delta$, for every $l,t$ and $\boldsymbol{p} \in \{\pm 1\}^2$ such that $\langle \boldsymbol{w}_l^{(t)}, \boldsymbol{p}\rangle > 0$, $(i, j, \widetilde{\boldsymbol{p}}) \notin \widetilde{\mathcal{P}}_\epsilon$ and $(l, \boldsymbol{p}) \notin A_t$, the following holds:*

$$\left(\sigma(\langle \boldsymbol{w}_l^{(t)}, \boldsymbol{p}\rangle) - \sigma(\langle \boldsymbol{w}_l^{(0)}, \boldsymbol{p}\rangle)\right) \cdot \widetilde{\gamma}(\boldsymbol{p})v_l\nu_j \ge 0$$

*Proof.* Notice that for every $t' \le t$ we have $(l, \boldsymbol{p}) \notin A_{t'} \subseteq A_t$. Therefore, using the previous lemma:

$$\left(\sigma(\langle \boldsymbol{w}_l^{(t)}, \boldsymbol{p}\rangle) - \sigma(\langle \boldsymbol{w}_l^{(0)}, \boldsymbol{p}\rangle)\right) \cdot \widetilde{\gamma}(\boldsymbol{p})v_l\nu_j = \sum_{1 \le t' \le t}\left(\sigma(\langle \boldsymbol{w}_l^{(t)}, \boldsymbol{p}\rangle) - \sigma(\langle \boldsymbol{w}_l^{(t')}, \boldsymbol{p}\rangle)\right) \cdot \widetilde{\gamma}(\boldsymbol{p})v_l\nu_j \ge 0$$

$\square$

Finally, we need to show that there are some "good" neurons, that are moving strictly away from zero:

**Lemma 12.** *Fix $\delta > 0$. Assume $\mathbb{P}_{(\boldsymbol{x},y)\sim\mathcal{D}}\left[\boldsymbol{x} \in \widetilde{\mathcal{X}}_{\epsilon'}\right] < \frac{\epsilon}{8\sqrt{2}n_i}\min\{\Delta, 2\beta\}$. Assume we sample $S \sim \mathcal{D}$, with $|S| > \frac{128}{\epsilon^2\min\{\Delta, 2\beta\}^2}\log\frac{4}{\delta}$. Assume that $k \ge \log^{-1}(\frac{4}{3})\log(\frac{4}{\delta})$. Then with probability at least $1 - 2\delta$, for every $\boldsymbol{p} \in \{\pm 1\}^2$ such that $(i, j, \widetilde{\boldsymbol{p}}) \notin \widetilde{\mathcal{P}}_\epsilon$, there exists $l \in [k]$ such that for every $t$ with $g_{t-1}(\boldsymbol{p}) \in (-1, 1)$, we have:*

$$\sigma(\langle \boldsymbol{w}_l^{(t)}, \boldsymbol{p}\rangle) \cdot \widetilde{\gamma}(\boldsymbol{p})v_l\nu_j \ge \eta t\frac{\epsilon}{\sqrt{2}n_i}\min\{\Delta, 2\beta\}$$

*Proof.* Assume the result of Lemma 9 holds (happens with probability at least $1 - \delta$). Fix some $\boldsymbol{p} \in \{\pm 1\}^2$ such that $(i, j, \widetilde{\boldsymbol{p}}) \notin \widetilde{\mathcal{P}}_\epsilon$. For $l \in [k]$, with probability $\frac{1}{4}$ we have both $v_l = \widetilde{\gamma}(\boldsymbol{p})\nu_j$ and $\langle \boldsymbol{w}_l^{(0)}, \boldsymbol{p}\rangle > 0$. Therefore, the probability that there exists $l \in [k]$ such that the above holds is $1 - (\frac{3}{4})^k \ge 1 - \frac{\delta}{4}$. Using the union bound, w.p at least $1 - \delta$, there exists such $l \in [k]$ for every $\boldsymbol{p} \in \{\pm 1\}^2$. In such case, we have $\langle \boldsymbol{w}_l^{(t)}, \boldsymbol{p}\rangle \ge \eta t\frac{\epsilon}{\sqrt{2}n_i}\min\{\Delta, 2\beta\}$, by induction:

- For $t = 0$ this is true since $\langle \boldsymbol{w}_l^{(0)}, \boldsymbol{p}\rangle > 0$.

- If the above holds for $t - 1$, then $\langle \boldsymbol{w}_l^{(t-1)}, \boldsymbol{p}\rangle > 0$, and therefore, using $v_l = \widetilde{\gamma}(\boldsymbol{p})\nu_j$ and Lemma 9:

$$-\langle\frac{\partial L_{\Psi(\mathcal{D})}}{\partial \boldsymbol{w}_l^{(t)}}, \boldsymbol{p}\rangle > \widetilde{\gamma}(\boldsymbol{p})v_l\nu_j\frac{\epsilon}{\sqrt{2}n_i}\min\{\Delta, 2\beta\}$$

And we get:

$$\begin{aligned}
\langle\boldsymbol{w}_l^{(t)}, \boldsymbol{p}\rangle &= \langle\boldsymbol{w}_l^{(t-1)}, \boldsymbol{p}\rangle - \eta\langle\frac{\partial L_{\Psi(\mathcal{D})}}{\partial \boldsymbol{w}_l^{(t)}}, \boldsymbol{p}\rangle \\
&> \langle\boldsymbol{w}_l^{(t-1)}, \boldsymbol{p}\rangle + \eta\widetilde{\gamma}(\boldsymbol{p})v_l\nu_j\frac{\epsilon}{\sqrt{2}n_i}\min\{\Delta, 2\beta\} \\
&\geq \eta(t-1)\frac{\epsilon}{\sqrt{2}n_i}\min\{\Delta, 2\beta\} + \eta\frac{\epsilon}{\sqrt{2}n_i}\min\{\Delta, 2\beta\}
\end{aligned}$$

$\square$

Using the above results, we can analyze the behavior of $g_t(\boldsymbol{p})$:

**Lemma 13.** *Assume we initialize $\boldsymbol{w}_l^{(0)}$ such that $\left\|\boldsymbol{w}_l^{(0)}\right\| \leq \frac{1}{4k}$. Fix $\delta > 0$. Assume $\mathbb{P}_{(\boldsymbol{x},y)\sim\mathcal{D}}\left[\boldsymbol{x} \in \widetilde{\mathcal{X}}_{\epsilon'}\right] < \frac{\epsilon}{8\sqrt{2}n_i}\min\{\Delta, 2\beta\}$. Assume we sample $S \sim \mathcal{D}$, with $|S| > \frac{128}{\epsilon^2\min\{\Delta,2\beta\}^2}\log\frac{4}{\delta}$. Assume that $k \geq \log^{-1}(\frac{4}{3})\log(\frac{4}{\delta})$. Then with probability at least $1 - 2\delta$, for every $\boldsymbol{p} \in \{\pm1\}^2$ such that $(i, j, \widehat{\boldsymbol{p}}) \notin \widetilde{\mathcal{P}}_\epsilon$, for $t > \frac{3n_i}{\sqrt{2}\eta\epsilon\min\{\Delta,2\beta\}}$ we have:*

$$g_t(\boldsymbol{p}) = \widetilde{\gamma}(\boldsymbol{p})\nu_j$$

*Proof.* Using Lemma 12, w.p at least $1 - 2\delta$, for every such $\boldsymbol{p}$ there exists $l_{\boldsymbol{p}} \in [k]$ such that for every $t$ with $g_{t-1}(\boldsymbol{p}) \in (-1, 1)$:

$$v_{l_{\boldsymbol{p}}}\sigma(\langle\boldsymbol{w}_{l_{\boldsymbol{p}}}^{(t)}, \boldsymbol{p}\rangle) \cdot \widetilde{\gamma}(\boldsymbol{p})\nu_j \geq \eta t\frac{\epsilon}{\sqrt{2}n_i}\min\{\Delta, 2\beta\}$$

Assume this holds, and fix some $\boldsymbol{p} \in \{\pm1\}^2$ with $(i, j, \widehat{\boldsymbol{p}}) \notin \widetilde{\mathcal{P}}_\epsilon$. Let $t$, such that $g_{t-1}(\boldsymbol{p}) \in (-1, 1)$. Denote the set of indexes $J = \{l : \langle\boldsymbol{w}_l^{(t)}, \boldsymbol{p}\rangle > 0\}$. We have the following:

$$\begin{aligned}
g_t(\boldsymbol{p}) &= \sum_{l\in J}v_l\sigma(\langle\boldsymbol{w}_l^{(t)}, \boldsymbol{p}\rangle) \\
&= v_{l_{\boldsymbol{p}}}\sigma(\langle\boldsymbol{w}_{l_{\boldsymbol{p}}}^{(t)}, \boldsymbol{p}\rangle) + \sum_{l\in J\backslash\{l_{\boldsymbol{p}}\},(l,\boldsymbol{p})\notin A_t}v_l\sigma(\langle\boldsymbol{w}_l^{(t)}, \boldsymbol{p}\rangle) + \sum_{l\in J\backslash\{l_{\boldsymbol{p}}\},(l,\boldsymbol{p})\in A_t}v_l\sigma(\langle\boldsymbol{w}_l^{(t)}, \boldsymbol{p}\rangle)
\end{aligned}$$

From Corollary 3 we have:

$$\widetilde{\gamma}(\boldsymbol{p})\nu_j \cdot \sum_{l\in J\backslash\{l_{\boldsymbol{p}}\},(l,\boldsymbol{p})\notin A_t}v_l\sigma(\langle\boldsymbol{w}_l^{(t)}, \boldsymbol{p}\rangle) \geq -k\sigma(\langle\boldsymbol{w}_l^{(0)}, \boldsymbol{p}\rangle) \geq -\frac{1}{4}$$

By definition of $A_t$ and by our assumption on $\eta$ we have:

$$\widetilde{\gamma}(\boldsymbol{p})\nu_j \cdot \sum_{l\in J\backslash\{l_{\boldsymbol{p}}\},(l,\boldsymbol{p})\in A_t}v_l\sigma(\langle\boldsymbol{w}_l^{(t)}, \boldsymbol{p}\rangle) \geq -k\frac{4\eta}{n_i} \geq -\frac{1}{4}$$

Therefore, we get:

$$\widetilde{\gamma}(\boldsymbol{p})\nu_j \cdot g_t(\boldsymbol{p}) \geq \eta t\frac{\epsilon}{\sqrt{2}n_i}\min\{\Delta, 2\beta\} - \frac{1}{2}$$

This shows that for $t > \frac{3n_i}{\sqrt{2}\eta\epsilon\min\{\Delta,2\beta\}}$ we get the required. $\square$

*Proof.* of Lemma 6. Using the result of Lemma 13, with union bound over all choices of $j \in [n_i/2]$. The required follows by the definition of $\widetilde{\gamma}(x_{2j-1}, x_{2j}) = \gamma_{i-1,j}(\xi_{2j-1}x_{2j-1}, \xi_{2j}x_{2j})$, and using the definition of $\widetilde{\mathcal{X}}_\epsilon$ $\square$

*Proof.* of Lemma 1. Fix some $i \in [d], j \in [n_i/2], \boldsymbol{p} \in \{\pm 1\}^2, y' \in \{\pm 1\}$, such that:

$$\mathbb{P}_{(\boldsymbol{x},y) \sim \mathcal{D}^{(i)}} \left[ \gamma_{i-1,j}(x_{2j-1}, x_{2j}) = \gamma_{i-1,j}(\boldsymbol{p}) \right] > 0$$

Assume w.l.o.g. that $j = 1$. Denote by $W$ the set of all possible choices for $x_3, \ldots, x_{n_i}$, such that when $(x_1, x_2) = \boldsymbol{p}$, the resulting label is $y'$. Formally:

$$W := \{(x_3, \ldots, x_{n_i}) \ : \ \Gamma_{i\ldots d}(p_1, p_2, x_3, \ldots, x_{n_i}) = y'\}$$

Then we get:

$$\begin{aligned}
&\mathbb{P}_{\mathcal{D}^{(i)}} \left[ (x_1, x_2) = \boldsymbol{p}, y = y', \gamma_{i-1,j}(x_1, x_2) = \gamma_{i-1,j}(\boldsymbol{p}) \right] \\
&= \mathbb{P}_{\mathcal{D}^{(i)}} \left[ (x_1, x_2) = \boldsymbol{p}, (x_3, \ldots, x_{n_i}) \in W, \gamma_{i-1,j}(x_1, x_2) = \gamma_{i-1,j}(\boldsymbol{p}) \right] \\
&= \mathbb{P}_{\mathcal{D}^{(i)}} \left[ (x_1, x_2) = \boldsymbol{p}, \gamma_{i-1,j}(x_1, x_2) = \gamma_{i-1,j}(\boldsymbol{p}) \right] \cdot \mathbb{P}_{\mathcal{D}^{(i)}} \left[ (x_3, \ldots, x_{n_i}) \in W \right] \\
&= \mathbb{P}_{\mathcal{D}^{(i)}} \left[ (x_1, x_2) = \boldsymbol{p} | \gamma_{i-1,j}(x_1, x_2) = \gamma_{i-1,j}(\boldsymbol{p}) \right] \cdot \mathbb{P}_{\mathcal{D}^{(i)}} \left[ \gamma_{i-1,j}(x_1, x_2) = \gamma_{i-1,j}(\boldsymbol{p}), (x_3, \ldots, x_{n_i}) \in W \right] \\
&= \mathbb{P}_{\mathcal{D}^{(i)}} \left[ (x_1, x_2) = \boldsymbol{p} | \gamma_{i-1,j}(x_1, x_2) = \gamma_{i-1,j}(\boldsymbol{p}) \right] \cdot \mathbb{P}_{\mathcal{D}^{(i)}} \left[ y = y', \gamma_{i-1,j}(x_1, x_2) = \gamma_{i-1,j}(\boldsymbol{p}) \right]
\end{aligned}$$

And dividing by $\mathbb{P}_{\mathcal{D}^{(i)}} \left[ \gamma_{i-1,j}(x_1, x_2) = \gamma_{i-1,j}(\boldsymbol{p}) \right]$ gives the required. □

## C   PROOFS OF SECTION 5

*Proof.* of Lemma 2.

For every gate $(i, j)$, let $J_{i,j}$ be the subset of leaves in the binary tree whose root is the node $(i, j)$. Namely, $J_{i,j} := \{(j-1)2^{d-i} + 1, \ldots, j2^{d-i}\}$. Then we show inductively that for an input $\boldsymbol{x} \in \{\pm 1\}^n$, the $(i, j)$ gate outputs: $\prod_{l \in I \cap J_{i,j}} x_l$:

- For $i = d - 1$, this is immediate from the definition of the gate $\gamma_{d-1,j}$.

- Assume the above is true for some $i$ and we will show this for $i - 1$. By definition of the circuit, the output of the $(i - 1, j)$ gate is a product of the output of its inputs from the previous layers, the gates $(i, 2j - 1), (i, 2j)$. By the inductive assumption, we get that the output of the $(i - 1, j)$ gate is therefore:

$$\left( \prod_{l \in J_{i,2j-1} \cap I} x_l \right) \cdot \left( \prod_{l \in J_{i,2j} \cap I} x_l \right) = \prod_{l \in (J_{i,2j-1} \cup J_{i,2j}) \cap I} x_l = \prod_{l \in J_{i-1,j}} x_l$$

From the above, the output of the target circuit is $\prod_{l \in J_{0,1} \cap I} x_l = \prod_{l \in I} x_l$, as required. □

*Proof.* of Lemma 3.

By definition we have:

$$c_{i,j} = \mathbb{E}_{(\boldsymbol{x},y) \sim \mathcal{D}} \left[ \Gamma_{(i+1)\ldots d}(\boldsymbol{x})_j y \right] = \mathbb{E}_{(\boldsymbol{x},y) \sim \mathcal{D}} \left[ \Gamma_{(i+1)\ldots d}(\boldsymbol{x})_j y \right] = \mathbb{E}_{(\boldsymbol{x},y) \sim \mathcal{D}} \left[ \Gamma_{(i+1)\ldots d}(\boldsymbol{x})_j x_1 \cdots x_k \right]$$

Since we require $\mathcal{I}_{i,j} \neq 0$, then we cannot have $\Gamma_{(i+1)\ldots d}(\boldsymbol{x})_j \equiv 1$. So, from what we showed previously, it follows that $\Gamma_{(i+1)\ldots d}(\boldsymbol{x})_j = \prod_{j' \in I'} x_{j'}$ for some $\emptyset \neq I' \subseteq I$. Therefore, we get that:

$$c_{i,j} = \mathbb{E}_{\mathcal{D}} \left[ \prod_{j' \in I \setminus I'} x_{j'} \right] = \prod_{j' \in I \setminus I'} \mathbb{E}_{\mathcal{D}} \left[ x_{j'} \right] = \prod_{j' \in I \setminus I'} (2p_{j'} - 1)$$

Furthermore, we have that:

$$\mathbb{E}_{\mathcal{D}} \left[ y \right] = \mathbb{E}_{\mathcal{D}} \left[ \prod_{j' \in I} x_{j'} \right] = \prod_{j' \in I} \mathbb{E}_{\mathcal{D}} \left[ x_{j'} \right] = \prod_{j' \in I} (2p_{j'} - 1)$$

And using the assumption on $p_j$ we get:

$$|c_{i,j}| - |\mathbb{E}_{\mathcal{D}}[y]| = \prod_{j' \in [k] \setminus I'} |2p_{j'} - 1| - \prod_{j' \in [k]} |2p_{j'} - 1|$$

$$= \left( \prod_{j' \in [k] \setminus I'} |2p_{j'} - 1| \right) \left( 1 - \prod_{j' \in I'} |2p_{j'} - 1| \right)$$

$$\geq \left( \prod_{j' \in [k] \setminus I'} |2p_{j'} - 1| \right) \left( 1 - (1 - 2\xi)^{|I'|} \right)$$

$$\geq (2\xi)^{k - |I'|} \left( 1 - (1 - 2\xi) \right) \geq (2\xi)^k$$

Now, for the second result, we have:

$$\mathbb{P}_{(z,y) \sim \Gamma_{i\ldots d}(\mathcal{D})}[z_j = 1] = \mathbb{E}_{(x,y) \sim \mathcal{D}} \left[ \mathbf{1}\{ \Gamma_{(i+1)\ldots d}(x)_j = 1 \} \right]$$

$$= \mathbb{E}_{(x,y) \sim \mathcal{D}} \left[ \frac{1}{2} (\prod_{j' \in I'} x_{j'} + 1) \right]$$

$$= \frac{1}{2} \prod_{j' \in I'} \mathbb{E}_{(x,y) \sim \mathcal{D}}[x_{j'}] + \frac{1}{2}$$

And so we get:

$$\left| \mathbb{P}_{(z,y) \sim \Gamma_{i\ldots d}(\mathcal{D})}[z_j = 1] - \frac{1}{2} \right| = \frac{1}{2} \prod_{j' \in I'} |\mathbb{E}_{(x,y) \sim \mathcal{D}}[x_{j'}]|$$

$$< \frac{1}{2}(1 - 2\xi)^{|I'|} \leq \frac{1}{2} - \xi$$

$\square$

*Proof.* of Lemma 4 For every $i \in [d]$ and $j \in [2^i]$, denote the following:

$$p_{i,j}^+ = \mathbb{P}_{(x,y) \sim \mathcal{D}^{(i)}}[x_j = 1 | y = 1], \ p_{i,j}^- = \mathbb{P}_{(x,y) \sim \mathcal{D}^{(i)}}[x_j = 1 | y = -1]$$

Denote $\mathcal{D}^{(i)}|_z$ the distribution $\mathcal{D}^{(i)}$ conditioned on some fixed value $z$ sampled from $\mathcal{D}^{(i-1)}$. We prove by induction on $i$ that $|p_{i,j}^+ - p_{i,j}^-| = \left( \frac{2}{3} \right)^i$:

- For $i = 0$ we have $p_{i,j}^+ = 1$ and $p_{i,j}^- = 0$, so the required holds.

- Assume the claim is true for $i - 1$, and notice that we have for every $z \in \{\pm 1\}^{2^{i-1}}$:

$$\mathbb{P}_{(x,y) \sim \mathcal{D}^{(i)}}[x_j = 1 | y = 1] = \mathbb{P}_{(x,y) \sim \mathcal{D}^{(i)}|_z}[x_j = 1 | z_{\lceil j/2 \rceil} = 1] \cdot \mathbb{P}_{(z,y) \sim \mathcal{D}^{(i-1)}}[z_{\lceil j/2 \rceil} = 1 | y = 1]$$
$$+ \mathbb{P}_{(x,y) \sim \mathcal{D}^{(i)}|_z}[x_j = 1 | z_{\lceil j/2 \rceil} = -1] \cdot \mathbb{P}_{(z,y) \sim \mathcal{D}^{(i-1)}}[z_{\lceil j/2 \rceil} = -1 | y = 1]$$

$$= \begin{cases} p_{i-1, \lceil j/2 \rceil}^+ + \frac{1}{3}(1 - p_{i-1, \lceil j/2 \rceil}^+) & if \ \gamma_{i-1, \lceil j/2 \rceil} = \wedge \\ \frac{2}{3} p_{i-1, \lceil j/2 \rceil}^+ & if \ \gamma_{i-1, \lceil j/2 \rceil} = \vee \\ \frac{1}{3} p_{i-1, \lceil j/2 \rceil}^+ + (1 - p_{i-1, \lceil j/2 \rceil}^+) & if \ \gamma_{i-1, \lceil j/2 \rceil} = \neg\wedge \\ \frac{2}{3}(1 - p_{i-1, \lceil j/2 \rceil}^+) & if \ \gamma_{i-1, \lceil j/2 \rceil} = \neg\vee \end{cases}$$

$$= \begin{cases} \frac{2}{3} p_{i-1, \lceil j/2 \rceil}^+ - \frac{1}{3} & if \ \gamma_{i-1, \lceil j/2 \rceil} = \wedge \\ \frac{2}{3} p_{i-1, \lceil j/2 \rceil}^+ & if \ \gamma_{i-1, \lceil j/2 \rceil} = \vee \\ 1 - \frac{2}{3} p_{i-1, \lceil j/2 \rceil}^+ & if \ \gamma_{i-1, \lceil j/2 \rceil} = \neg\wedge \\ \frac{2}{3} - \frac{2}{3} p_{i-1, \lceil j/2 \rceil}^+ & if \ \gamma_{i-1, \lceil j/2 \rceil} = \neg\vee \end{cases}$$

Similarly, we get that:

$$\mathbb{P}_{(\boldsymbol{x},y)\sim\mathcal{D}^{(i)}}\left[x_j = 1 | y = -1\right] = \begin{cases} \frac{2}{3}p^-_{i-1,\lceil j/2\rceil} - \frac{1}{3} & if\ \gamma_{i-1,\lceil j/2\rceil} = \wedge \\ \frac{2}{3}p^-_{i-1,\lceil j/2\rceil} & if\ \gamma_{i-1,\lceil j/2\rceil} = \vee \\ 1 - \frac{2}{3}p^-_{i-1,\lceil j/2\rceil} & if\ \gamma_{i-1,\lceil j/2\rceil} = \neg\wedge \\ \frac{2}{3} - \frac{2}{3}p^-_{i-1,\lceil j/2\rceil} & if\ \gamma_{i-1,\lceil j/2\rceil} = \neg\vee \end{cases}$$

Therefore, we get:

$$|p^+_{i,j} - p^-_{i,j}| = \frac{2}{3}|p^+_{i-1,\lceil j/2\rceil} - p^-_{i-1,\lceil j/2\rceil}| = \left(\frac{2}{3}\right)^i$$

From this, we get:

$$\begin{aligned} \left|\mathbb{E}_{(\boldsymbol{x},y)\sim\mathcal{D}^{(i)}}\left[x_j y\right]\right| &= \left|\mathbb{E}_{(\boldsymbol{x},y)\sim\mathcal{D}^{(i)}}\left[(2\mathbf{1}\{x_j = 1\} - 1)y\right]\right| \\ &= \left|2\mathbb{E}_{(\boldsymbol{x},y)\sim\mathcal{D}^{(i)}}\left[\mathbf{1}\{x_j = 1\}y\right] - \mathbb{E}\left[y\right]\right| \\ &= \left|2\left(\mathbb{P}_{\mathcal{D}^{(i)}}\left[x_j = 1, y = 1\right] - \mathbb{P}_{\mathcal{D}^{(i)}}\left[x_j = 1, y = -1\right]\right) - \mathbb{E}\left[y\right]\right| \\ &= \left|2\left(p^+_{i,j}\mathbb{P}\left[y = 1\right] - p^-_{i,j}\mathbb{P}\left[y = -1\right]\right) - \mathbb{E}\left[y\right]\right| \\ &= \left|2\left(\frac{1}{2}(p^+_{i,j} - p^-_{i,j}) + \xi(p^+_{i,j} + p^-_{i,j})\right) - \mathbb{E}\left[y\right]\right| \\ &\geq \left|p^+_{i,j} - p^-_{i,j}\right| - 2\xi\left|p^+_{i,j} + p^-_{i,j}\right| - |\mathbb{E}\left[y\right]| \\ &\geq \left|p^+_{i,j} - p^-_{i,j}\right| - 6\xi > \frac{1}{2}\left(\frac{2}{3}\right)^d \end{aligned}$$

And hence:

$$\left|\mathbb{E}_{(\boldsymbol{x},y)\sim\mathcal{D}^{(i)}}\left[x_j y\right]\right| - \left|\mathbb{E}_{(\boldsymbol{x},y)\sim\mathcal{D}^{(i)}}\left[y\right]\right| \geq \frac{1}{2}\left(\frac{2}{3}\right)^d - 2\xi > \frac{1}{3}\left(\frac{2}{3}\right)^d$$

$\square$

*Proof.* of Lemma 5 Fix some $\boldsymbol{z}' \in \{\pm 1\}^{n_i/2}$ and $y' \in \{\pm 1\}$. Then we have:

$$\begin{aligned} \mathbb{P}_{(\boldsymbol{x},y)\sim\Gamma_i(\mathcal{D}^{(i)})}\left[(\boldsymbol{x}, y) = (\boldsymbol{z}', y')\right] &= \mathbb{P}_{(\boldsymbol{x},y)\sim\mathcal{D}^{(i)}}\left[(\Gamma_i(\boldsymbol{x}), y) = (\boldsymbol{z}', y')\right] \\ &= \mathbb{P}_{(\boldsymbol{x},y)\sim\mathcal{D}^{(i)}}\left[\forall j\ \gamma_{i-1,j}(x_{2j-1}, x_{2j}) = z'_j\ and\ y = y'\right] \\ &= \mathbb{P}_{(\boldsymbol{z},y)\sim\mathcal{D}^{(i-1)}}\left[(\boldsymbol{z}, y) = (\boldsymbol{z}', y')\right] \end{aligned}$$

By the definitions of $\mathcal{D}^{(i)}$ and $\mathcal{D}^{(i-1)}$.

$\square$

