# OpenReview forum: "Learning Boolean Circuits with Neural Networks"
_ICLR.cc/2020/Conference — Reject_

### Official Review · AnonReviewer2 · 2019-10-17
**Official Blind Review #2**

**Rating:** 3

**Review:**


Summary:
The paper proposes a layer-wise method for training the weights of a binary-tree-structured neural network such that it correctly reproduces certain classes of Boolean functions defined by binary-tree-structured Boolean circuits. Specifically, this paper shows analytically that if a circuit satisfies a property termed “local correlation” where there is sufficient correlation between every gate in the circuit and the true output label of the circuit, then this circuit can be learned by a neural network with the same structure as the circuit by training it one layer at a time from the input to the output. The paper motivates this by showing empirically that the k-parity problem with some bias to the labels can be learned by a neural network, but that this does not work when there is no bias in the labels, implying that this bias is necessary for successful learning. The paper shows formally that instances of the k-parity problem satisfy the local correlation assumption and can thus be learned, and also shows that there exists at least one distribution given by a simple generative model that satisfies this assumption and is thus also learnable in this manner.

Overall:
Weak reject. My main concerns are as follows.

First, if you define a circuit such that each gate’s output is correlated with the actual label, and then simply copy that structure and learn each gating function independently to predict the value that it is correlated with, then it seems likely that each additional gate should improve the representation. And because the network mimics exactly the circuit structure, the capacity of the network will not be a problem either. Thus, while it’s not trivial to say that the function can be recovered exactly, it is not exactly surprising either, and I don’t see what this offers in terms of novel intuition.

Second, the definitions and assumptions are not very reflective of what is actually done in deep learning, and I do not see a clear connection that would make this result useful in the field. It doesn’t connect well enough to actual datasets, problems, models, or learning algorithms that are used, so I do not see what insights can be taken from it.

However, I do not have a problem with the quality of the paper, and think it would find a more appropriate audience at a different venue.

Clarity: The paper is quite well written and intelligible, although the notation is fairly dense and not always intuitive (e.g., numbering layers of the network from output to input).
Significance: I do not think the results are particularly significant.

Detailed comments:
Section 1.
- The claims made in this section come off overly strong for the remainder of the paper. For example, training a deep neural network is not necessarily computationally hard because training is almost always taken to mean using SGD to locally minimize some loss function, and not taken to mean global optimization. This should be made more clear. Further, in my opinion, the “holy grail” of theoretical deep learning right now is the work on understanding why these networks generalize, not on global optimization or exact function learning (although there is some overlap between these two goals).
- Can you better explain why correlation between the input bits and the label being necessary is not obviously necessary for all standard correlation-based learning methods? I agree that it would be nice if our methods were not so limited but this seems to simply reiterate the fact that they are. (e.g., I would not expect a deep network to learn on ImageNet if there was no correlation between some pixels and the output).
- The ImageNet example is not particularly clear. First, the appendix implies you are taking center crops from each image, not random crops as stated in the intro. Second, I assume that you take a center crop per image, but the text is written to sound like only a single crop for all of ImageNet is used, which doesn’t make sense.
- I do not think that the ImageNet experiment implies that local correlation exists in ImageNet any more than is already well known. It simply shows that an even smaller center crop from ImageNet images is still mildly predictive of the output classes; however, since ImageNet images are generally object-centered already, this is not particularly surprising.

Section 2.
- This is a well-written section that does a good job of situating this work.

Section 3.
- The assumption that the circuit must be a binary tree is quite restrictive and excludes pretty much all common deep learning architectures. Could this be relaxed empirically, even if not analytically?
- In neural networks, it is common to define layer 1 as the input and layer d as the output, and it wasn’t immediately clear from the notation here that the opposite was being done here. Please make this more clear early on (or flip the ordering).
- In the neural-gate definition, should v_i be v_l? Otherwise what is i indexing? Further, since v is not actually learned later, can it just be removed? This would simplify the notation somewhat.
- How are v and w initialized?

Section 4.
- While it is strong to assume that every gate’s output should correlate with the label, it’s natural to think that many of each layer’s output in an actual neural network will correlate with the output, since earlier layers can be thought of as an input corresponding to a transformed representation of the actual input. This is also implied by the Belilovsky et al. layer-wise training work. Making these points more clearly and providing some empirical evidence could strengthen this paper’s claims. Perhaps measure the correlation between outputs in different layers and the labels?
- Assumption 2 is at odds with the fact that datasets are typically designed to be as balanced as possible. Can you reconcile these?

Appendix.
- It would be nice if this read as well as the main paper.


**Experience Assessment:**

I do not know much about this area.

**Review Assessment: Checking Correctness Of Derivations And Theory:**

I assessed the sensibility of the derivations and theory.

**Review Assessment: Checking Correctness Of Experiments:**

I carefully checked the experiments.

**Review Assessment: Thoroughness In Paper Reading:**

I read the paper at least twice and used my best judgement in assessing the paper.

---

> ### Author Response · Authors · 2019-11-14
> **Response to Reviewer #2**
>
> We thank you for your comments and feedback.
>
> As a general note, we believe that the fact that our theoretical results are aligned with common intuition is a good sign. However, while many heuristic algorithms follow some intuitions that we believe makes them work, proving that they indeed work is a very fundamental problem in understanding such algorithms.
>
> As for the specific comments:
> Section 1:
> - We agree that the word "train" is somewhat misleading. A better phrasing for our claim is that neural networks are hard to LEARN. That is, it is hard to efficiently find a neural-network that performs well on an unseen sample, given a finite sample from the distribution. Note that our results show that the layer-wise optimization can learn the distributions discussed in the paper, which means that we get both good optimization and good generalization.
> - The fact that our result is aligned with our intuition about the behavior of the algorithm is good. However, showing rigorously that local correlation is necessary and sufficient, in some cases, for the convergence of the algorithm is not trivial, and as far as we know was never shown in previous theoretical works.
> - There was a mistake in the description of the experiment in the appendix. The network gets a single patch chosen randomly from the image, and not a center crop as stated in the appendix. This was fixed in a new version we uploaded.
>
> Section 3:
> - Any theoretical result in the literature requires some assumption on the target function, thus limiting the possible target functions for which the results apply. Many works assume more restrictive assumptions, for example - assuming that the target function is linear or approximated by a large-margin function in a kernel space. It is hard to give a result that applies to all functions that can be calculated by a deep network, as this family of functions includes essentially all functions that can be efficiently computed.
> - v is initialized randomly, with values in {1,-1}, and w is initialized from spherically symmetric distribution satisfying the assumptions stated in the theorems.
>
> Section 4:
> - Our experiment shows that even a single patch has some correlation to the target function. Since an output of a neuron in higher layers of the network integrates information from multiple patches, it can be expected that this output will be correlated with the label. As the primary focus of the paper is on theoretical analysis, we included only a few limited experiments to motivate our theoretical results.
> - We assume that the label is slightly biased, since in this case the neural-gates corresponding to nodes with no influence simply converge to the bias of the label, and hence get "stuck" on a constant function, sending all patterns to a single bit (the sign of the bias). When there is no bias, the gradient on the distribution for variables with no influence is zero, and hence the convergence is governed by the stochastic noise from the empirical sample. We believe that the same result can be given when there is no label bias, using a more complicated analysis, but chose to analyze the simpler case.

---

> > ### Comment · AnonReviewer2 · 2019-11-15
> > **Response**
> >
> > Having read the author response and re-read parts of the paper, let me try to clarify my thoughts on this paper.
> >
> > 1.
> > (a) In the abstract, “local correlation” is defined as correlation between the input and the target label (call this type A local correlation) but is later more explicitly defined to mean correlation between all gates in the network and the target label (call this type B local correlation). This discrepancy is somewhat jarring as the latter seems much stronger than the first. In my opinion, the paper does a poor job of properly separating these, and the conflation of the two leads to difficulties in determining the value of this contribution.
> >
> > (b) In particular, the latter definition of local correlation (type B) comes off as (essentially) assuming the exact result to be shown. If each gate is correlated with the label, then a simple linear classifier on the output gates is learnable and sufficient to learn the function. It seems like it should be possible to start with the type A definition and show that there are cases where the latter definition follows from it, making the main result much stronger.
> >
> > (c) While perhaps no theoretical results exist that build on local correlation (in either type), focusing only on theoretical work ignores the wide body of empirical research that likely exploits this property (in the type A sense). If existing networks and training methods do exploit this obvious-seeming property, then this should be made clear. If they don’t, then the paper should explain why this intuition is incorrect.
> >
> >
> > 2.
> > (a) The paper is written as if correlation between inputs and labels being useful is an unexpected discovery and a contribution of this paper. However, local correlation (type A) seems trivially obvious in most real datasets, such as the ImageNet example provided. The (likely) widespread existence of local correlation in most real datasets makes it seem like a weak condition, which is good. However, if it is weak, it should be easy to show that it exists, which is not done, thus not sufficiently justifying the contributions.
> >
> > (b) That said, it’s not clear that local correlation is a weak condition for datasets of Boolean functions. It may actually be a fairly strong condition for these datasets. This ambiguity between whether this is a strong or weak assumption, and the conflation between Boolean function datasets and datasets like ImageNet only muddy the argument and claims being made. Why not use binarized MNIST with two classes, or something similar that mimics the types of data and functions being discussed in this paper?
> >
> >
> > 3. In that regard, the ImageNet experiment comes off as somewhat misleading about the existence of local correlation. Given that ImageNet is fully observed, why not just measure the correlation between image patches and labels? Showing that a (non-linear) relationship exists between image patches and the labels (via a 2-layer network with ReLUs) does not necessarily show that correlation exists. It just shows that deep neural networks (that bear no resemblance to the architectures discussed in this paper) can learn a little bit of information on patches of ImageNet, which is not very surprising. This further exacerbates the discrepancy in my point (1) above, as this example speaks only of correlation between inputs and labels, and not about correlation between hidden layer gates and labels. A reasonable experiment here would be to simply perform linear regression between the input patches and the labels, allowing one to measure the correlation. This could also be done between the gates in a multi-layer network and the labels.
> >
> > I do not think that this paper is ready for publication in its current form and thus maintain my rating of Weak Reject.

---

### Official Review · AnonReviewer4 · 2019-10-20
**Official Blind Review #4**

**Rating:** 6

**Review:**

This paper aims to study the correlation between the neural network's input and output by abstracting the network as a binary tree Boolean circuit problem. The paper is well-written, motivations are clearly presented, and literature reviews are well placed. The contributions are mainly theoretical, and the experimental plots are simply used for concept illustrations, therefore the correctness of the theoretical analysis has no empirical evaluations.

Due to the rareness of the study regarding deep learning theory, this manuscript makes one step further towards understanding the training hardness on a few certain types of neural networks with different underlying distributions. Multiple strong assumptions have made to favor the analysis, however, due to the lack of expertise, I can understand parts of the intuitions behinds them. To mimic the Boolean circuits network, the authors have focused the analysis on a layerwise gradient-based training, which might be a potential drawback because modern deep models are much more complex (e.g., the Residual network architecture shares connections between layers) and this over-simplified analysis may be too restricted. My understanding is that this paper could become a very solid work if a few rules of thumb or intuitions can be proposed to guide finding the correlation property in realistic neural network architectures such that we can verify these proposed theories with some experiments.

Overall, I believe this is a fine theoretical foundation paper that should attract the attention of the researchers in deep learning community.

**Experience Assessment:**

I have read many papers in this area.

**Review Assessment: Checking Correctness Of Derivations And Theory:**

I assessed the sensibility of the derivations and theory.

**Review Assessment: Checking Correctness Of Experiments:**

I assessed the sensibility of the experiments.

**Review Assessment: Thoroughness In Paper Reading:**

I read the paper at least twice and used my best judgement in assessing the paper.

---

### Official Review · AnonReviewer1 · 2019-10-22
**Official Blind Review #1**

**Rating:** 6

**Review:**

Thank the authors for their rebuttal. It resolves all my previous concerns.
##############################

This paper proposes to use neural networks for learning binary tree structured boolean circuits. For the boolean circuits problem, the authors notice two importance factors influencing why the target circuit is easy to learn or not. The two factors include "local correlation" and "label bias". On the one hand, "local correlation" requires every influential node in the circuit have strong correlations with the target label, which makes the network trainable to exploit this correlation for minimizing losses. On the other hand,  "label bias" requires that there are not the same amount of positive and negative examples. The paper proves that the proposed algorithm can faithfully learn the target circuit  and presents multiple examples for the scenario of learning binary tree structured boolean circuits.

Strengths,
1, This paper points out the two key factors "local correlation" and "label bias" for the learnability of a boolean circuit. Empirically in Figure1, they also validates the finding by demonstrating problems with balanced labels are more difficult to train.
2, The paper puts their theoretical findings to the setup of k-parity problem, proving that their proposed algorithm can faithfully tackle the k-parity problem.

Weakness,
1, The main theorem proves that the target networks can be approximated using O(n^logn) examples. However, it it apparent that binary tree boolean circuits cannot represent all 2^n n-ary boolean functions. Actually, I GUESS all functions a binary tree can represent MIGHT also be in the scale of O(n^logn). Then the result is not surprising, as the training set probably covers all training examples. I think it is necessary that the authors give some estimates on the total number of representable functions and compare it with their |S|.
2, It is not clear to me why the the "label bias" is important for learning a boolean circuit. Could the authors clarify to me ?
3, It is also helpful to empirically compare the cases when there are local correlations and there aren't.
4, Sec 5.1 shows that the proposed algorithm can solve the k-parity problem, however they assumed that all parity nodes locates together. In practice, these parity nodes might scatter everywhere, which could incur big problems for a binary-tree  to approximate.

**Experience Assessment:**

I do not know much about this area.

**Review Assessment: Checking Correctness Of Derivations And Theory:**

I assessed the sensibility of the derivations and theory.

**Review Assessment: Checking Correctness Of Experiments:**

I carefully checked the experiments.

**Review Assessment: Thoroughness In Paper Reading:**

I read the paper at least twice and used my best judgement in assessing the paper.

---

> ### Author Response · Authors · 2019-11-14
> **Response to Reviewer #1**
>
> We thank you for your comment and feedback.
>
> 1. The number of functions calculated by a binary tree Boolean circuit is in fact exponential in the input dimension n, and grows like 6^n. An exact formula for the number of functions calculated by such tree is given in: https://arxiv.org/pdf/1904.02309.pdf. We fixed the paper, and added a clarification about this point. Indeed, the number of functions implemented by a tree structured Boolean circuit is much smaller than the total number of binary functions on n bits, which is 2^(2^n), but still there is a large number of functions in this class.
>
> 2. We assume that the label is slightly biased, since in this case the neural-gates corresponding to nodes with no influence simply converge to the bias of the label, and hence get "stuck" on a constant function, sending all patterns to a single bit (the sign of the bias). When there is no bias, the gradient on the distribution for variables with no influence is zero, and hence the convergence is governed by the stochastic noise from the empirical sample. We believe that the same result can be given when there is no label bias, using a more complicated analysis, but chose to analyze the simpler case.
>
> 3. The k-parity experiment shows the effect of local correlation on the behavior of the network. As the primary focus of the paper is on theoretical analysis, we included only a few limited experiments to motivate our theoretical results.
>
> 4. For simplicity, we showed the proof when the first k bits are the parity bits. It is a simple construction, in a similar fashion, to show that such circuit can express any k-parity. We fixed the paper to include the more general construction.

---

### Author Response · Authors · 2019-11-14
**Response to Reviewers**

We thank the reviewers for their feedback and comments. We posted separate responses answering the specific concerns raised by the reviewers. As a general comment, we find it unfortunate that two out of three reviewers are not familiar with the area of research discussed in the paper, as admitted by the reviewers themselves. We believe that reviewers with more experience in the field of theoretical machine learning would be able to provide feedback that better reflects the quality of this paper.

---

### Decision · Program_Chairs · 2019-12-19

**Decision:**

Reject

**Comment:**

The paper puts forward a theoretical investigation of the learnability of
tree-structured Boolean circuits with neural networks.
The authors identify *local correlations*, ie correlation of each internal
target circuit gate with the target output, as critical property for
characterizing learnability by layerwise training.

The reviewers agree that the paper is well written and content to be correct
(to the best of their knowledge).
However, they have reservations about the strength of the assumptions about the
target functions as well as the layerwise training procedure.

I think this paper is slightly below acceptance threshold for ICLR, which is a quite
applied conference.
The assumptions are quite strong, ie local correlations and the topology of the
circuit to be known as well as layerwise training, and possibly too far removed
from current deep learning practice.